# Tyrosine phosphorylation of the GARU E3 ubiquitin ligase promotes gibberellin signalling by preventing GID1 degradation

Keiichirou Nemoto[1], Abdelaziz Ramadan[1,2], Gen-ichiro Arimura[3], Kenichiro Imai[4], Kentaro Tomii [4], Kazuo Shinozaki[5] & Tatsuya Sawasaki[1]

Gibberellin (GA) is a major hormone for plant growth and development. GA response is derived from the degradation of DELLA repressor proteins after GA-dependent complex formation of the GID1 GA receptor with DELLA. Genistein is a known tyrosine (Tyr) kinase inhibitor and inhibits DELLA degradation. However, the biological role of Tyr phosphorylation on the GA response remains unclear. Here, we demonstrate that GARU (GA receptor RING E3 ubiquitin ligase) mediates ubiquitin-dependent degradation of GID1, and that the TAGK2 plant Tyr-kinase is a target of genistein and inhibits GARU–GID1A interactions by phosphorylation of GARU at Tyr321. Genistein induces degradation of GID1 and accumulation of DELLA. Conversely, *Arabidopsis garu* mutant and TAGK2-overexpressing plants accelerate GID1 stabilization and DELLA degradation. Under salt stress, GARU suppresses seed germination. We propose that GA response is negatively regulated by GARU-dependent GID1 ubiquitination and positively by Tyr phosphorylation of GARU by TAGK2, and genistein inhibits GA signaling by TAGK2 inhibition.

[1] Proteo-Science Center (PROS), Ehime University, 3 Bunkyo-cho, Matsuyama, Ehime 790-8577, Japan. [2] Faculty of Science, Botany Department, Ain Shams University, Cairo 11566, Egypt. [3] Faculty of Industrial Science & Technology, Department of Biological Science & Technology, Tokyo University of Science, Tokyo 125-8585, Japan. [4] Artificial Intelligence Research Center (AIRC) and Biotechnology Research Institute for Drug Discovery, National Institute of Advanced Industrial Science and Technology (AIST), 2-4-7 Aomi, Koto Ward, Tokyo 135-0064, Japan. [5] Gene Discovery Research Group, RIKEN Center for Sustainable Resource Science, 1-7-22 Suehiro-cho, Tsurumi-ku, Yokohama, Kanagawa 230-0045, Japan. Correspondence and requests for materials should be addressed to T.S. (email: sawasaki@ehime-u.ac.jp)

The phytohormone gibberellins (GAs) are diterpene compounds that control a wide range of growth and development[1]. The initiation of GA signaling involves four components: GA, the GA-receptor GID1 (GA INSENSITIVE DWARF1), the master repressor DELLA, and specific F-box protein[2]. GID1 was first identified in rice[3] and orthologous genes have been identified in a wide range of higher plants[4]. *Arabidopsis thaliana* has three homologous GID1 genes: GID1A, GID1B, and GID1C[5]. These may control the GA signaling pathway while being functionally redundant[5]. In *Arabidopsis*, five DELLA proteins have been identified: GAI, RGA, RGL1, RGL2, and RGL3. DELLA proteins mainly localize to the nucleus and act as negative or positive transcription regulators by interactions with specific transcription factors[6–8]. The first step in GA signaling involves GA binding to GID1. The GID1–GA complex then induces a complex formation with DELLA. The GID1–GA–DELLA complex interacts with an F-box protein that recruits the SCF E3 ligase complex for ubiquitination and the subsequent degradation of DELLA by the 26S proteasome. The rapid decrease of DELLA proteins causes a change in the activity of transcription factors and expression levels of several GA-regulated genes are either increased or decreased in plants.

Recent studies have shown that the stability of DELLA proteins is controlled by post-translational modification, such as serine/threonine phosphorylation/dephosphorylation of DELLA[9, 10]. In contrast, post-translational modifications of GID1 have yet to be reported. Interestingly, several pharmacological studies revealed that the broad Tyr kinase inhibitor genistein (GNS) strongly inhibited GA-induced DELLA degradation in barley and tobacco BY-2 cells[11, 12]. However, mutational analysis has shown that Tyr phosphorylation does not occur in DELLA proteins[12]. These results suggest that Tyr phosphorylation of GID1 and/or other proteins are involved in DELLA degradation and are critical in GA-induced DELLA degradation.

GNS is one of the predominant isoflavones in soybeans[13]. It is known to be an inhibitor of mammalian Tyr protein kinases (PTKs)[14, 15]. Although GNS is also found in many plant species, including *Arabidopsis*[16], its target molecule in plants remains unclear. A recent study showed that Tyr residue of CjWRKY1 is phosphorylated by unknown endogenous protein kinase(s) in *Coptis japonica* and its phosphorylation is inhibited by GNS treatment[17], suggesting that plants have protein kinase(s) targets of GNS. However, it is unclear whether Tyr phosphorylation signaling cascades occur in plants, because no PTK homologous genes have been found in *Arabidopsis* and rice genomes[18, 19]. Recently, several research groups have identified specific Tyr phosphatases in plants[20]. Tyr-phosphorylated peptides have been found by using a phosphoproteomic approach, and the proportion of Tyr phosphorylation observed was equivalent to that found in human cells[21]. These findings strongly suggest that plants have a Tyr phosphorylation signal pathway; although the role of Tyr phosphorylation in biochemical and physiological processes is poorly understood. In a previous study, we identified the angiosperm-specific CRK (calcium-dependent protein kinase-related protein kinase) family for Tyr phosphorylation[22]. CRKs could phosphorylate Tyr residues of beta-tubulin and certain transcription factors both in vitro and in plants. By genetic and biochemical analysis, it has been suggested that some CRKs are involved in the signal transduction of GA signaling, ABA signaling, floral development, and environmental stresses in *Arabidopsis* and tobacco[23, 24]. These findings suggest that Tyr phosphorylation by CRKs plays an important role in the signal pathways of the GA or ABA in plants.

In this study, we uncovered a molecular mechanism of how the stability of GA-receptor GID1 is negatively regulated by ubiquitination and positively regulated by Tyr phosphorylation, which is inhibited by GNS. Using a biochemical approach based on a wheat cell-free system, we identified an E3 ubiquitin ligase for the GA-receptor GID1, GARU (GA receptor RING E3 ubiquitin ligase), and its protein kinase TAGK2/CRK2 (renamed CRK2 TAGK2 because it is a target of GNS) for Tyr phosphorylation. Biochemical and genetic analysis revealed that GARU functions as a negative regulator of GA signaling in seedlings and seeds by inducing ubiquitin-dependent proteolysis of GID1s. However, Tyr321 of GARU was phosphorylated by TAGK2, resulting in a decrease in the accessibility to GID1A. TAGK2-dependent trans-phosphorylation of specific substrates ERF13 and GARU was inhibited by GNS in vitro and in cells. In addition, GNS treatment induced the destabilization of GID1s, but overexpression of *TAGK2* gene enhanced GID1s stability. These results suggested that TAGK2 plays a role of positive regulator for GA signaling by inactivation of GARU. Our key finding is therefore that GARU and TAGK2 regulate the GA signaling through regulating GID1 protein level.

## Results

**Promotion and degradation of GA receptor GID1**. Recent studies have shown that GNS inhibited GA-induced degradation of DELLA in barley and tobacco BY-2 cells[11, 12]. These results suggest that PTK is involved as a positive regulator of GA signaling through DELLA degradation in plants. Thus, we investigated the effect of GNS on the stability of DELLA and GID1 proteins in *Arabidopsis* seedlings. GNS treatment inhibited hypocotyl elongation and primary root growth in a dose-dependent manner (Fig. 1a). However, hypocotyl elongation of the quintuple *della* mutant (*gai-t6 rga-t2 rgl1-1 rgl3-1 rgl2-1/SGT625-5*) was not inhibited by GNS (Supplementary Fig. 1), suggesting that GNS-induced growth inhibition was in a DELLA activity-dependent manner. In addition, endogenous RGA protein levels were increased by GNS treatment (Fig. 1b). In contrast, the GID1A/C protein level was slightly decreased by GNS treatment and significantly decreased by GNS treatment with cycloheximide (CHX), which is a protein synthesis inhibitor (Fig. 1c). These results suggest that GNS induces the instability of GID1 proteins and consequently provides stabilization for DELLA proteins.

Next, we investigated whether the GNS-induced decrease of GID1 protein level was a result of proteasomal degradation. The C-terminus of GID1A was fused with an AGIA-tag with an epitope sequence specifically recognized by a high-affinity rabbit monoclonal antibody[25] and GID1A-AGIA protein was expressed in *Arabidopsis* protoplasts, using a transient expression system. Similar to the endogenous GID1 in Fig. 1c, exogenous GID1A-AGIA level was decreased by GNS treatment (GNS in Fig. 1d) and, in contrast, treatments of gibberellin (GA₃) and proteasome inhibitor (MG132) stabilized it. The GNS-induced decrease of GID1A level was also partially rescued by supplementation with MG132 (GNS + MG132). The anti-AGIA antibody detected GID1A-AGIA and high molecular weight smear bands (>80 kDa) were detected with long exposure (Fig. 1d, lane Mock). Furthermore, the intensities of the high molecular weight smear bands were increased by treatment with GNS and GNS + MG132. These results suggest that the GID1A protein is ubiquitinated and degraded by proteasomes, GNS enhances the ubiquitination of GID1A protein, and GA treatment inhibits GID1A ubiquitination.

**GA receptor RING E3 ubiquitin ligase**. Recently, we made an *Arabidopsis* RING-type E3 ligase array consisting of 204 E3 ligases[26] with a high-throughput screening method[27]. Thus, we have attempted to identify E3 ubiquitin ligases for ubiquitination

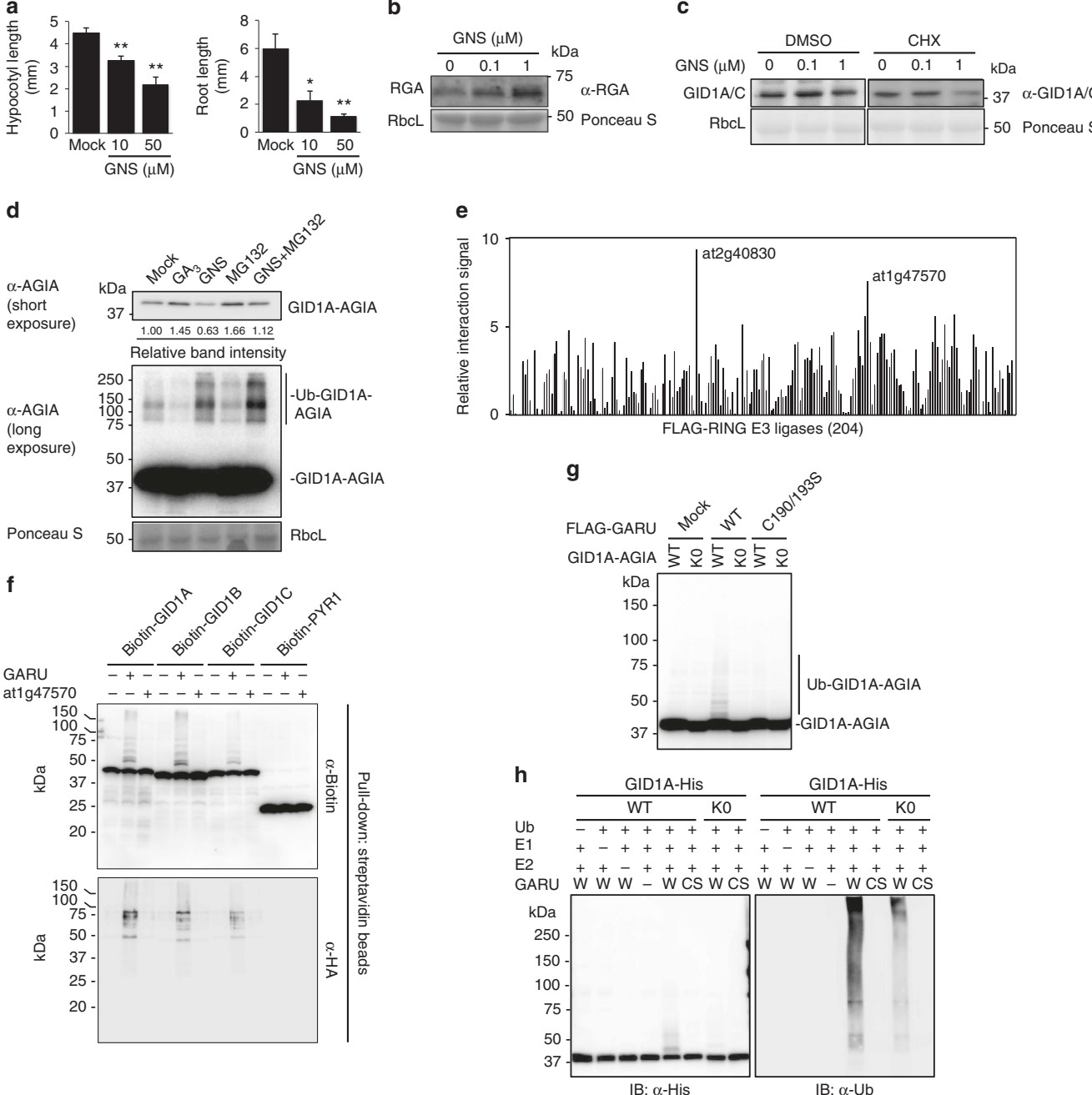

**Fig. 1** Identification of GID1-interacting RING-type E3 ubiquitin ligase GARU. **a** Hypocotyl elongation assay of etiolated seedlings (left) and primary root elongation assay (right) in response to GNS. Error bars indicate ±s.d. of three replicates, $n = 24$ (hypocotyl), and $n = 20$ (root). Statistically significant changes compared with Mock are indicated (*$P < 0.05$, **$P < 0.01$, as determined using two-tailed Student's $t$-test). **b** Immunoblot analysis of RGA protein in Col-0 seedlings that were treated with GNS for 6 h. **c** Immunoblot analysis of endogenous GID1A/C proteins in Col-0 seedlings that were treated with GNS and 300 μM CHX for 6 h. **d** Immunoblot analysis of GID1A-AGIA in protoplasts that were treated with 10 μM GA$_3$, 5 μM GNS, or 20 μM MG132 for 6 h. Relative protein level was calculated as the signal intensity ratio of the Mock. **e** PPI analysis of biotin-GID1A and 204 FLAG-E3s. **f** In vitro ubiquitination assay using the top two interacting clones. Biotin-GID1A, GID1B, and GID1C were incubated with recombinant HA-tagged ubiquitin and E2 in the presence (+) or absence (−) of FLAG-tagged GARU or at1g47579 protein. After incubation, reaction mixture was boiled in 2% SDS, and biotin-GID1 proteins were collected by affinity purification using streptavidin magnetic beads. Biotin-GID1 proteins and HA-ubiquitin were detected by immunoblot analysis using anti-biotin antibody or anti-HA antibody. PYL1 was negative control. **g** In vitro ubiquitination assay using GID1A$^{K0}$ mutant. C-terminal AGIA-tagged GID1A (WT) or GID1A$^{K0}$ mutant (K0) was incubated with FLAG-GARU (WT) or FLAG-GARU$^{C190/193S}$ mutant (C190/193S). Ubiquitinated GID1A (Ub-GID1A-AGIA) proteins were detected by immunoblot analysis using anti-AGIA antibody. **h** In vitro ubiquitination assay using purified GID1A and GARU. Affinity purified GID1A-His or GID1A$^{K0}$-His was incubated in the presence (+) or absence (−) of ubiquitin (Ub), recombinant E1, E2, wild-type FLAG-GARU (W), and FALG-GARU$^{C190/193S}$ mutant (CS). GID1A-His and ubiquitinated proteins were detected by immunoblotting using anti-His antibody or anti-ubiquitin antibody

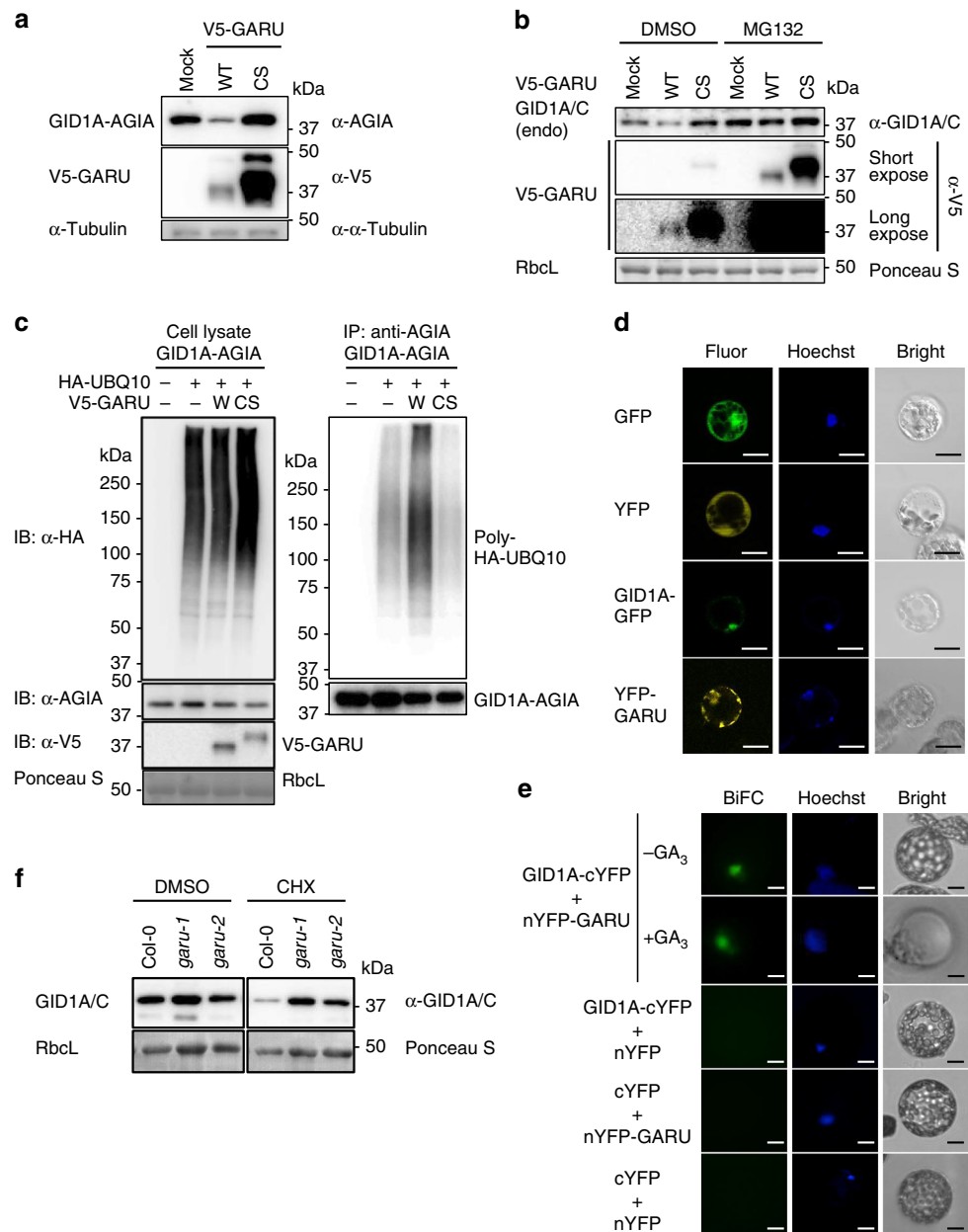

**Fig. 2** GARU-dependent ubiquitination induces degradation of GID1s. **a** Destabilization assay of exogenous GID1A-AGIA. GID1A-AGIA was co-expressed with V5-GARU (WT) or V5-GARU$^{C190/193S}$ mutant (CS) in protoplasts. **b** Destabilization assay of endogenous GID1. V5-GARU (WT) or V5-GARU$^{C190/193S}$ mutant (CS) was transfected into protoplasts. After incubation with 20 μM MG132 or 0.1% DMSO for 6 h, expressed proteins were analyzed by immunoblotting. **c** In cell ubiquitination assay of GID1A. GID1A-AGIA was co-expressed with V5-GARU or V5-GARU$^{C190/193S}$ (CS), and 3 × (HA-UBQ10) in protoplasts. After incubation with 20 μM MG132 for 6 h, GID1A-AGIA was immunoprecipitated with anti-AGIA antibody sepharose beads from protoplasts lysates. Ubiquitinated proteins (HA-UBQ10) were detected by immunoblotting using anti-HA antibody. **d** Subcellular localizations of GID1A-GFP and YFP-GARU in *Arabidopsis* protoplasts. Fluor, GFP, or YFP fluorescence image; Hoechst, Hoechst 33258 stain fluorescence image; Bright, bright-field image. Scale bars = 20 μm. **e** BiFC analysis of GARU–GID1A interaction in *Arabidopsis* protoplast. The C-terminal half of YFP fused GID1A (GID1A-cYFP) and the N-terminal half of YFP fused GARU (nYFP-GARU) were co-expressed in protoplasts, and incubated with or without of 10 μM GA$_3$. nYFP and cYFP were co-expressed with GID1A-cYFP or nYFP-GARU as a negative controls. BiFC, YFP fluorescence image; Hoechst, Hoechst 33258 stain fluorescence image; Bright, bright-field image. Scale bars = 10 μm. **f** Immunoblot analysis of GID1 proteins in Col-0, *garu-1*, and *garu-2* that were treated with 0.1% DMSO or 300 μM CHX for 20 h

of GID1A, using a wheat cell-free system and AlphaScreen technology. Using this approach, we identified two RING-type E3 ubiquitin ligases, at2g40830 and at1g47570, which interacted strongly with GID1A (Fig. 1e and Supplementary Table 1). Subsequently, to confirm ubiquitination activity with GID1A, we carried out in vitro ubiquitination assays without purification using the two candidate E3s. The at2g40830 protein ubiquitinated

not only GID1A, but also GID1B and GID1C in vitro (Fig. 1f), whereas the at1g47570 protein did not show activity with any GID1 proteins, and we therefore named at2g40830 GARU (GA receptor RING E3 ubiquitin ligase). To investigate whether GID1A ubiquitination is dependent on the RING domain (a region of 190–231 a.a.) of GARU and occurs on the lysine residue in GID1A, we made two mutants, a CS mutant of GARU (C190/

193S) with each cysteine changed to serine and a lysine-lacking GID1A mutant (K0) in which all lysine residues were changed to arginine. The CS mutant of GARU did not ubiquitinate GID1A and the GID1A$^{K0}$ mutant was not ubiquitinated by GARU (Fig. 1g), indicating that ubiquitination of GID1A is dependent on the RING domain in GARU and is carried out at the lysine residue (s) in GID1A. To confirm this, all the proteins were purified and used for the in vitro ubiquitination assay. Similar to the cell-free-based ubiquitination assay, GID1A protein was ubiquitinated by GARU proteins (Fig. 1h). These results suggest that GARU interacts with and ubiquitinates GID1 proteins in vitro. Since 17 lysine residues were found in the amino acid sequence of GID1A, we prepared single, double, and quadruple arginine substitution variants and carried out in vitro ubiquitination assays to identify ubiquitination site(s). However, all of these mutants were ubiquitinated by GARU (Supplementary Fig. 2), suggesting that its ubiquitination occurs on multiple lysine residues.

**Proteolytic degradation of GID1 by GARU.** Since one of the major roles of ubiquitination is the degradation of the target protein by 26S proteasome, we expected that GARU might induce degradation of GID1A after ubiquitination in the cells. To test this hypothesis, we co-transfected GARU and GID1A into *Arabidopsis* protoplast and investigated the GARU-dependent destabilization of GID1A. The expression level of exogenous GID1A-AGIA was significantly decreased with a wild-type (WT) form of GARU (WT in Fig. 2a), whereas co-expression with CS mutant (CS) showed the same level as the control (Mock). Furthermore, endogenous GID1A/C level was also decreased by the expression of WT GARU, but not the CS mutant, and the GARU-dependent decrease of GID1A/C protein level was almost completely rescued by treatment with MG132 proteasome inhibitor (Fig. 2b), suggesting that the decrease of GID1 protein level by GARU is a result of ubiquitin-dependent protein degradation by the proteasome. To analyze GARU-dependent GID1A ubiquitination in the cells, WT GARU or its inactive form mutant was co-transfected with GID1A-AGIA and HA-UBQ10 into the protoplast. Cell lysates were immunoprecipitated with an anti-AGIA antibody to recover GID1A-AGIA proteins and subsequently polyubiquitination of GID1A-AGIA was detected with the anti-HA antibody. Similar to in vitro polyubiquitination of GID1 protein shown in Fig. 1h, the levels of polyubiquitinated GID1A-AGIA increased with expression of WT GARU (W lane in right panel of Fig. 2c), but not with the CS mutant. Taken together, these results suggest that GID1 protein is degraded by the GARU-dependent ubiquitination in plant cells.

Next, we examined the localization of the GARU protein in *Arabidopsis* cells. Transient expression analyses revealed that YFP-GARU is partially localized in the nucleus and GID1A-GFP is mainly localized in the nucleus of cells (Fig. 2d). Furthermore, bimolecular fluorescent complementation (BiFC) analysis indicated that YFP fluorescence was observed in the nucleus in the presence and absence of GA$_3$ (Fig. 2e), indicating that GARU could interact with GID1 with or without GA$_3$ in the nucleus.

To investigate the GARU-dependent stability of GID1A/C protein in plants, we used the two T-DNA tagged lines, *garu-1* (SALK_037121C) and *garu-2* (SALK_127503C) (Supplementary Fig. 3). Expression levels of GID1A/C protein in the mutants were almost the same as that in WT Col-0 plants under normal conditions (left panel in Fig. 2f). However, proteins stability assay using CHX showed that endogenous GID1A/C protein level was more dramatically stabilized in both *garu* mutants than in WT Col-0 plants (right panel). Taken together, these results suggest that GARU induces ubiquitin-dependent degradation of GID1A in the nucleus of plant cells.

**TAGKs/CRKs are a target of GNS.** DELLA degradation is suppressed by treatment with GNS (Fig. 1b). Although GNS (Fig. 3a) is known as an inhibitor of mammalian PTKs[15], the target of GNS in plants is unknown. Furthermore, several studies have shown that Tyr residues of beta-tubulins are phosphorylated in plants[28], and GNS and Tyr-phosphatase inhibitors affect microtubule organization[29, 30]. In our previous study, we reported that Tyr residues of beta-tubulin 2 and 7 were phosphorylated by CRK2 and CRK3 in plants[22]. On the basis of these findings, we speculated that CRKs might be a target of GNS. To test this hypothesis, we investigated whether the activity of the CRKs was inhibited by GNS. Treatment with GNS dramatically inhibited the activity of CRK2/TAGK2 and CRK3/TAGK3 at very low levels (IC$_{50}$ = 2.56 and 3.95 nM, respectively, Fig. 3b), whereas GNS analogs did not (Supplementary Fig. 4a, b), and hereafter we call CRKs TAGKs (target of GNS kinases). GNS also inhibited the activity of GmTAGK3 at a very low concentrations (Supplementary Fig. 4c), although CDPK3/CPK3 and MKK9 were not affected (Supplementary Fig. 5), indicating a high specificity for TAGK proteins. In addition, the TAGK inhibition by GNS was observed in cells (Fig. 3c). To understand the mechanism of action of GNS, we generated 3D models of the AtTAGK3 kinase domain interacting with GNS, based on the domain structure of nine protein kinases (Supplementary Figs. 4d, e, 6). These results suggest that GNS is a highly sensitive inhibitor binding to the active site of TAGK proteins.

**Tyr phosphorylation of GARU by TAGK2.** From the two findings that GNS treatment also suppressed expression of GID1 proteins (Fig. 1c, d) and inhibited TAGK activity (Fig. 3b, c), we investigated whether GARU was a substrate of TAGK. First, conserved Tyr residues were screened in GARU orthologs among dicot and monocot plants because GNS has been reported to suppress GA signaling in many plant species[11, 12]. GARU has a conserved N-terminal zinc-finger motif and C-terminal RING-H2 motif for ubiquitination activity and a characteristic poly-serine motif at the C-terminal (Fig. 4a). Multiple sequence alignment of homologous GARU gene sequences revealed that a conserved Tyr residue at the C-terminal region was found between orthologous sequences in *Arabidopsis*, *Z. mays*, *O. sativa*, *G. max*, *H. vulgare*, and *S. lycopersicum* (Fig. 4a). Therefore, we prepared a C-terminal deletion GARU mutant (ΔC) and a single amino acid substitution mutant (Y321F; Tyr321 was replaced with Phe) and subsequently analyzed whether the ΔC mutant and its conserved Tyr321 residue are phosphorylated by TAGK2 or TAGK3. An in vitro kinase assay showed that only TAGK2 was able to phosphorylate WT GARU (Fig. 4b) and TAGK2-dependent Tyr phosphorylation was not observed in GARU-ΔC and Y321F mutants (Fig. 4c). Similar analysis was performed with mutants of other Tyr residues, but a difference was not observed between the wild type and mutants (Supplementary Fig. 7).

To confirm TAGK2-dependent Tyr phosphorylation of GARU in the cells, we performed in cell phosphorylation assays using a protoplast transient expression system using the *TAGK2*-deficient mutant to avoid the effects of endogenous TAGK2. Since the WT form of exogenous GARU was expressed at a very low level (Fig. 2a), we used the GARU$^{CS}$ mutant instead. In addition, a phospho-protein mobility shift assay with Phos-tag acrylamide[22, 31] was used to detect the phosphorylated GARU protein. In the Phos-tag SDS-PAGE assay, in which the phosphorylated proteins bound to the divalent metal ions decreases the migration speed, mobility-shifted GARU$^{CS}$ bands were observed in a TAGK2 activity-dependent manner (Phos-GARU in Fig. 4d); however, the band for GARU$^{Y321F}$ was barely detectable. Next, we examined the effect of GNS on TAGK2-dependent

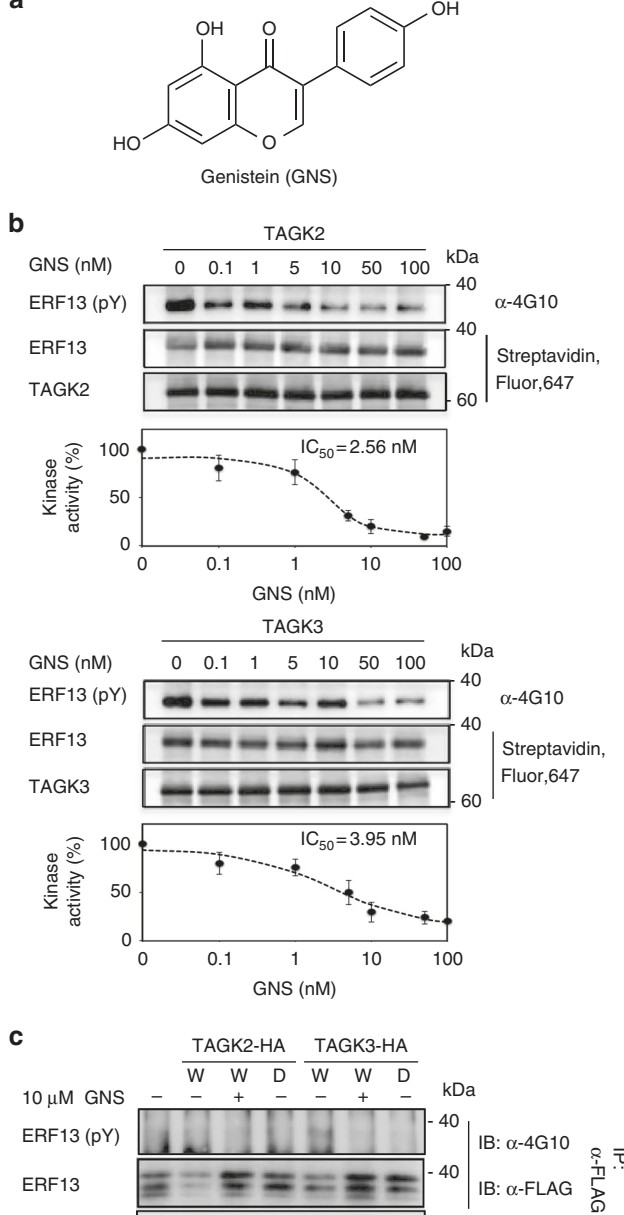

**Fig. 3** TAGK activities are inhibited by GNS. **a** Structures of GNS. **b** In vitro inhibition assay of TAGKs by GNS. Tyr phosphorylation (pY) was detected by anti-phospho-Tyr antibody (4G10) (upper panel). Concentration-inhibition curves for GNS were quantified from at least three independent in vitro kinase assays (lower panel). Data are means (±s.d.) of three independent experiments ($n = 3$). ERF13 is substrate for TAGK2 and TAGK3. **c** Tyr phosphorylation analysis of ERF13 in protoplasts. FLAG-ERF13 was co-expressed with TAGK2-HA or TAGK3-HA (wild-type, W; kinase dead, D) in protoplasts. After incubation with 10 μM GNS or 0.1% DMSO (control) for 24 h, FLAG-ERF13 was immunoprecipitated using an anti-FLAG antibody agarose beads form protoplasts lysates. Tyr phosphorylation (pY) was detected by 4G10 antibody. Expression of TAGK2-HA and TAGK3-HA was detected by anti-HA antibody

phosphorylation of GARU using a protoplast derived from WT ecotype Col-0. In the Phos-tag SDS-PAGE, the mobility-shifted bands were detected by the expression of GARU$^{CS}$ alone (left lane in Fig. 4e), indicating that GARU was phosphorylated by endogenous TAGK2. In addition, the mobility-shifted GARU$^{CS}$

bands were increased by co-expression with TAGK2 (middle lane) and were significantly decreased by a GNS-treatment (right lane). These results indicate that TAGK2 could phosphorylate at Tyr321 of GARU in cells and its phosphorylation is inhibited by GNS.

**TAGK2 inhibits GARU-dependent degradation of GID1.** Next, we investigated the influence of Tyr phosphorylation of GARU on an interaction between GARU and GID1 because the two proteins were found to bind to each other (Fig. 1e). Interestingly, in vitro co-immunoprecipitation of recombinant proteins using anti-V5 antibodies showed that supplementation with active-form TAGK2 (W) decreased interactions between HA-tagged GARU and V5-tagged GID1A (Fig. 4f), whereas TAGK3 and an inactive form of TAGK2 (D) were unaffected, suggesting that Tyr phosphorylation of GARU inhibits binding to GID1 proteins. Immunoprecipitation of V5-PYR1 as a negative control did not show interaction with HA-GARU, indicating that the pull-down result of HA-GARU is a specific interaction between HA-GARU and GID1A-V5. To quantitatively analyze the effect of Tyr phosphorylation of GARU on binding of GID1A further using AlphaScreen, Tyr at position 321 was substituted for other amino acids, such as Y321D or Y321E for a phosphorylation-mimic form and Y321F for a Tyr phosphorylation-null form. The C-terminal deletion mutant (ΔC) was also used for interaction detection. AlphaScreen analysis (Fig. 4g) revealed that GARU–GID1A interactions were suppressed in Tyr phosphorylation competent forms (lane 2: WT GARU and WT TAGK2), the ΔC mutant of GARU, and the two phosphorylation-mimic forms (Y321D and Y321E). In contrast, the phosphorylation-null form (Y321F) increased its interaction. These results suggest that the C-terminal region of GARU is important for interactions between GARU and GID1A and its interaction is inhibited by GARU phosphorylation at Tyr321 by TAGK2.

To observe the stabilization of GID1A under the control of Tyr321 phosphorylation of GARU, GARU and three mutants were expressed with GID1A in cells. Compared with WT GARU (Fig. 4h), the two phosphorylation-mimic forms of Y321D and Y321E partially rescued GID1A expression. To analyze the ubiquitination of the GID1 protein by Tyr321 phosphorylation of GARU in the cells, GID1A, GARU, and HA-UBQ10 were delivered into the protoplast cells. Ubiquitination of the GID1A protein was detected as an upper shift band (Ub-GID1A-AGIA) without co-expression of GARU. With co-expression with GARU, ubiquitination of GID1A was increased (mono-Ub-GID1A-AGIA) (WT in Fig. 4i). However, ubiquitination of GID1A decreased in the phosphorylation-mimic form (Y321E) of GARU, suggesting that TAGK2-dependent Tyr321 phosphorylation of GARU inhibits ubiquitination of GID1 protein. Next, we performed GNS-induced destabilization of GID1A protein in a TAGK2-deficient mutant (*tagk2*), transgenic plants overexpressing the WT TAGK2 gene (TAGK2$^{WT}$-OX1) or kinase-dead (KD) form (TAGK2$^{KD}$-OX1), and the *garu-1* mutant. In all transgenics and mutants, transcripts levels of GID1A gene were slightly decreased compared with WT Col-0 (Supplementary Fig. 8). However, in the presence of both CHX and GNS, GNS-induced reduction of GID1s protein was observed in the Col-0, *tagk2*, and TAGK2$^{KD}$-OX1, but was not observed in TAGK2$^{WT}$-OX1 and *garu-1* mutants (Fig. 4j). These results suggested that GNS-dependent destabilization of GID1s level occurs in a TAGK2 and GARU activity-dependent manner.

**GA blocks GARU-dependent GID1A degradation.** In this study, we found that GNS, but not its analog, can inhibit TAGK2 kinase activity (Fig. 3b and Supplementary Fig. 4b). Thus, we tested

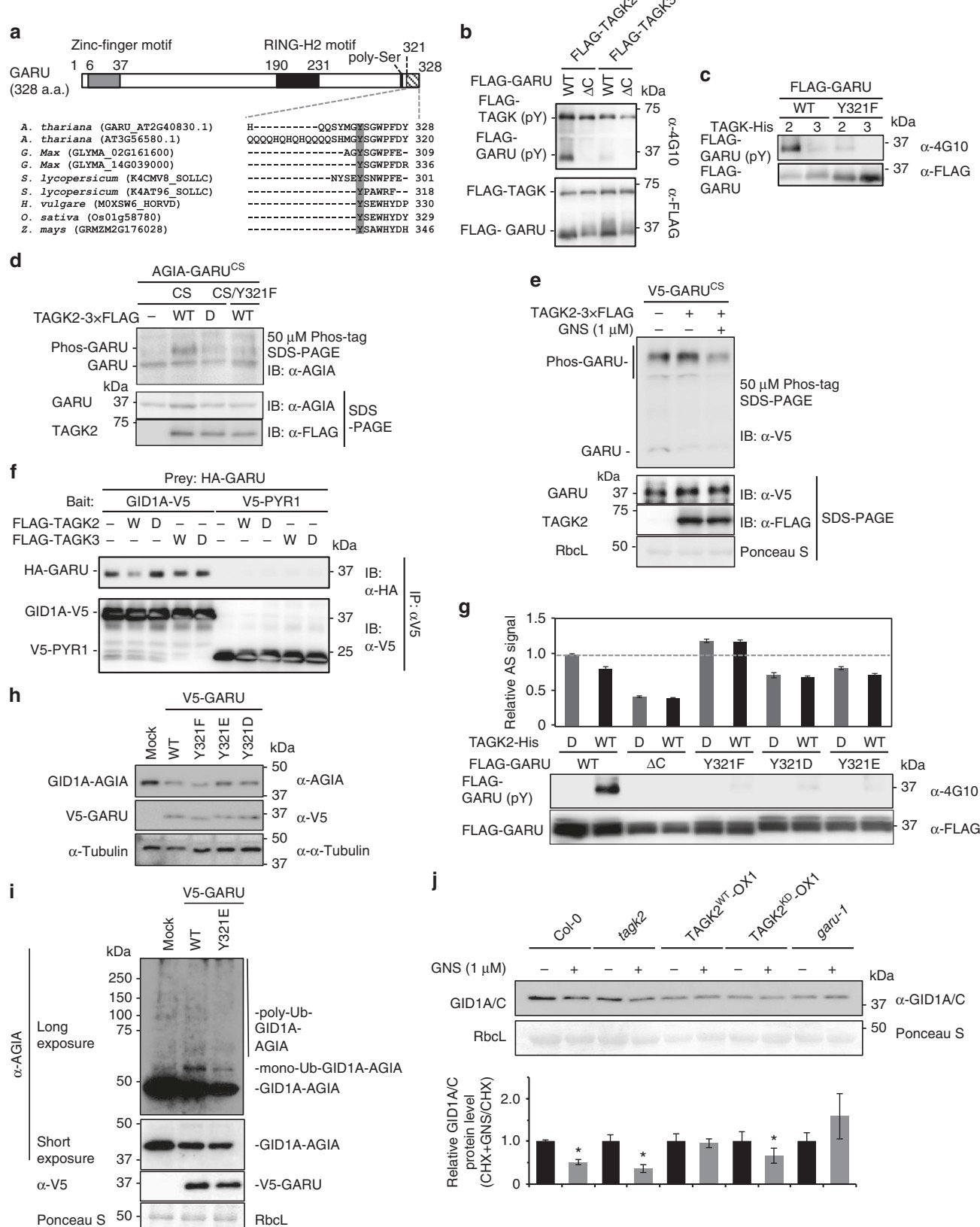

whether GNS-induced growth arrest is the GNS-specific effect and if it is recovered by treatment with exogenous GA. A hypocotyl elongation assay of etiolated seedlings showed that growth arrest was observed in a GNS treatment, but not in the presence of daidzein (Fig. 5a). In addition, GNS-induced growth arrest was recovered by treatment with exogenous GA₃ (Fig. 5a). In the presence of GNS, although the DELLA protein level was increased (see also Fig. 1b), the level of GA-induced DELLA

(RGA) degradation after 60 min was almost the same with or without GNS treatment (Fig. 5b). GA is known to induce a complex formation between GID1A and DELLA[5]. Therefore, we next investigated whether GARU can ubiquitinate a GID1–GA–DELLA complex. To detect a GA-dependent GID1A–GAI interaction, an assay method based on AlphaScreen technology was constructed (Supplementary Fig. 9). GARU was able to interact with the GID1A–GA–DELLA complex and did not affect GID1A–GA–DELLA complex formation (Fig. 5c, d). However, GARU-dependent GID1A ubiquitination decreased in the presence of DELLA protein, RGL2 (Fig. 5e), suggesting that GARU cannot mediate ubiquitination of GID1A in the GID1A–GA–DELLA complex. Furthermore, GNS-induced GID1A ubiquitination was greatly reduced by supplementation with GA₃ in the cells (Fig. 5f). These results suggest that GID1 protein in the GA response does not play a role on the GARU substrate even though functional GARU level increases by GNS-induced TAGK2 inhibition. We therefore concluded that the GA response is more predominant than GARU-dependent GID1 ubiquitination.

### GARU and TAGK2 mediate sensitivity of GA response

To investigate the roles of GARU and TAGK2 on GA response, we monitored the levels of endogenous RGA in the seedling of the *tagk2* mutants, TAGK2^WT-OXs, TAGK2^KD-OX1, and the *garu-1* mutant. No significant differences in RGA protein levels were detected in all plants (Fig. 6a). However, when these plants were treated with GA, degradation of RGA was enhanced in the *garu-1* mutant and TAGK2^WT-OX1 plant (Fig. 6b, c). These results suggest that *garu* deficit or *TAGK2*-overexpression induces the increased degradation of DELLA by stabilizing GID1A. Next, we compared the growth responses of hypocotyl in the dark among these transgenic and mutant plants with or without exogenous GA₃. To avoid the effects of endogenous GAs, we tested the GA sensitivity in the presence of paclobutrazol (PAC) as a GA biosynthesis inhibitor. PAC treatment of WT and all transgenic plants reduced hypocotyl elongation by approximately 40% compared with untreated plants (Fig. 6d). In contrast, after being treated with different concentrations of GA₃, the hypocotyl lengths of the *garu-1* mutant and TAGK2^WT-OX1 plants dramatically increased. The differences in sensitivity to GA treatments among these plants correlated well with their DELLA protein levels (Fig. 6b, c). Taken together, these results suggest that GARU and TAGK2 mediate sensitivity of GA response in *Arabidopsis* plants.

### Salt stress-dependent GID1 degradation in the seeds

Next, we analyzed the biological role of GARU. According to the *Arabidopsis* eFP Browser[32], gene expression level of *GARU* increases at the late dry maturation stages, and decreases after imbibition of water, indicating that it may regulate seed maturation, dormancy, or germination. To investigate this hypothesis, we examined GID1s protein level in the seeds. Although GID1s protein was not observed in dry after-ripened seeds, it was found after water imbibition in Col-0 (Fig. 7a). In contrast, in the *garu-1* mutant, accumulation of GID1s protein was observed in both dry after-ripened seeds and imbibed seeds. Furthermore, the *garu-1* mutant germinated earlier than the Col-0 (Fig. 7b). These results suggest that GID1s protein is degraded by GARU in the stage of seed maturation and GARU negatively regulates seed germination.

Expression level of GID1s protein in Col-0 seeds is dramatically increased by water imbibition, and ABA, GA, or PAC treatment and both low and high temperature did not influence its expression (Fig. 7c). However, GID1s protein decreased following salt treatment (Fig. 7c), which was almost completely rescued by treatment with MG132 (Fig. 7d). In *garu-1* mutant seeds, the salt-mediated decrease of GID1s protein was not observed (Fig. 7e). In the condition of salt with exogenous GA₃, although the salt-mediated decrease of GID1s protein in Col-0 was greatly suppressed, it was not significant changed in *garu-1* mutant seeds (Fig. 7f). Gene expression analysis revealed that transcripts of *GID1A*, *GID1B*, and *GID1C* did not change with treatment of water and/or salt and that one of *GARU* decreased by half after water imbibition and its expression was maintained equal to that of the dry seeds by treatment with salt solution (Fig. 7g). Furthermore, germination assay under salt stress condition revealed that the germination capacity of *garu-1* seeds was higher than that observed in the Col-0 seeds, and germination of *garu-1* was dramatically increased by exogenous GA₃ treatment (Fig. 7h). Taken together, these results suggest that GARU functions as a suppressor of seed germination under salt stress through the degradation of GID1 protein.

### Discussion

All of the results in this study indicate that GID1s are ubiquitinated by GARU, and Tyr phosphorylation of GARU by TAGK2 and TAGK2 inhibition by GNS mediate GA response in *Arabidopsis* plants. Furthermore, in seeds, salt stress induces the gene expression of GARU, and GARU-dependent destabilization of GID1s protein causes decreases in GA sensitivity and germination. On the basis of our results, we proposed the biochemical and biological roles of GARU, TAGK2, and GNS in the GA signaling (Fig. 8).

**Fig. 4** TAGK2-dependent Tyr phosphorylation of GARU represses GARU–GID1A interaction. **a** Diagram of the structure of GARU (upper) and multiple sequence alignment of GARU protein in different species (lower). **b**, **c** In vitro kinase assay of GARU and its mutants by TAGK2 and TAGK3. GARU (WT) and GARU-ΔC mutant (ΔC) (**b**) or GARU^Y321F mutant (Y321F) (**c**). Tyr phosphorylation (pY) was detected by immunoblotting using anti-phospho-Tyr antibody (4G10). **d**, **e** In cell phosphorylation analysis of GARU mutants by Phos-tag SDS-PAGE. AGIA-GARU^C190/193S mutant (CS) or AGIA-GARU^C190/193S/Y321F mutant (CS/Y321F) was co-expressed with TAGK2-3 × FLAG (wild-type, WT; kinase dead, D) in *tagk2* protoplasts (**d**). V5-GARU^C190/193S mutant (CS) was co-expressed with TAGK2-3 × FLAG in the presence (+) or absence (−) of 1 μM GNS (**e**). **f** GARU–GID1A interaction analysis by pull-down assay. HA-GARU (prey) was incubated with FLAG-TAGK2 or TAGK3 (wild-type, W; kinase dead, D), and then incubated with GID1A or PYR1 (bait). V5-tagged proteins were immunoprecipitated (IP) with anti-V5 antibody, and analyzed by immunoblotting (IB) using anti-HA antibody. **g** GARU–GID1A interaction analysis by AlphaScreen. FLAG-GARU and its mutants (ΔC, Y321F, Y321D, and Y321E) were incubated with TAGK2-His (wild-type, WT; kinase dead, D), and then incubated with biotin-GID1A. GARU–GID1A interaction was analyzed by AlphaScreen (upper), and Tyr phosphorylation (pY) of GARU was analyzed by immunoblotting using anti-phospho Tyr antibody (4G10, lower). Relative AlphaScreen (AS) signal was quantified as the PPI signal value ratio of FLAG-GARU^WT (pre-incubated with TAGK2^KD)-biotin-GID1A. Data are means (±s.d.) of three independent experiments (*n* = 3). **h**, **i** In cell co-expression analysis of GID1A-AGIA with V5-GARU (WT) or its mutants (Y321F, Y321E, and Y321D) in the absence (**h**) or presence (**i**) of MG132. **j** Immunoblot analysis of endogenous GID1A/C in Col-0, *tagk2*, TAGK2^WT-OX1, TAGK2^KD-OX1, and *garu-1*. Seedlings were treated with 300 μM CHX and 1 μM GNS. Relative GID1A/C protein level was calculated as the signal intensity ratio of the without GNS treatment. Data are means (±s.d.) of four independent experiments (*n* = 4). Statistically significant changes compared with in the absence of GNS are indicated (*$P < 0.05$, as determined using two-tailed Student's *t*-test)

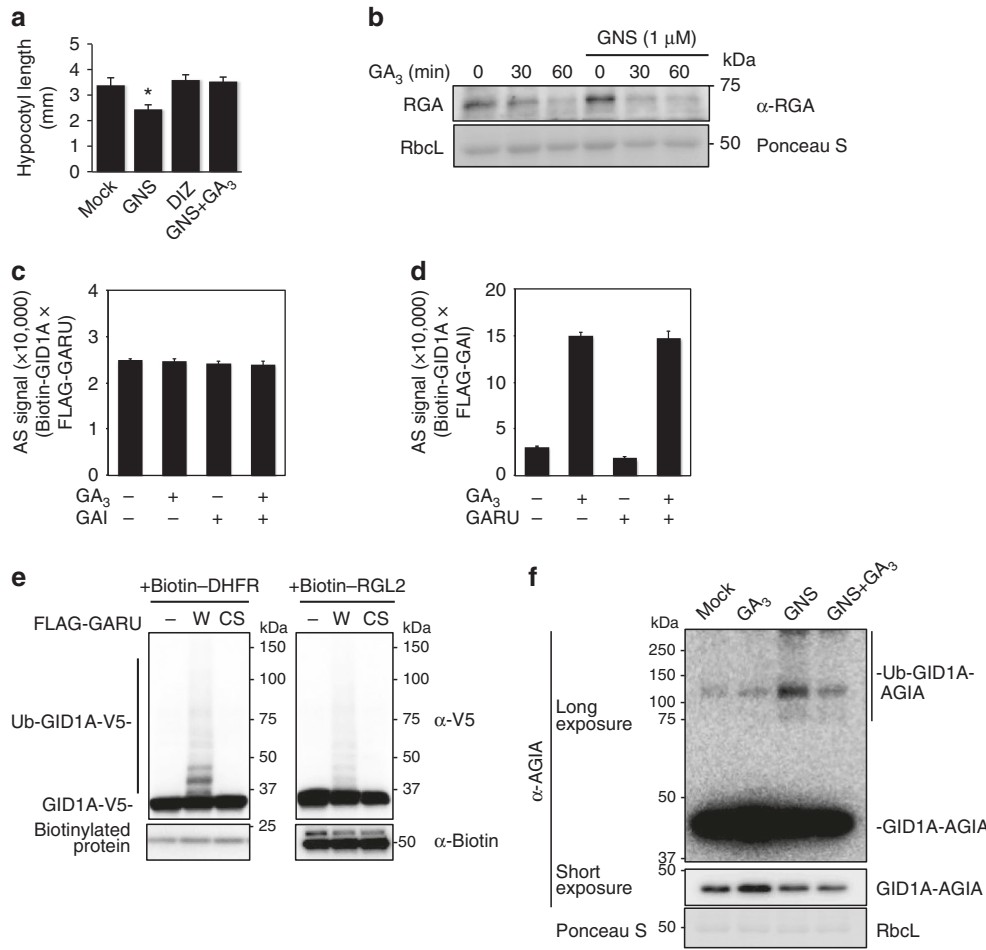

**Fig. 5** GA blocks GARU-dependent GID1A ubiquitination. **a** Hypocotyl elongation assay of etiolated seedlings in response to 10 μM GNS, 10 μM daidzein (DIZ), and 10 μM GNS with 5 μM GA$_3$ (GNS + GA$_3$). Error bars indicate ±s.d. of three replicates, $n = 21$. Statistically significant changes compared with Mock are indicated (*$P < 0.05$, as determined using two-tailed Student's $t$-test). **b** GA-induced RGA degradation assay in Col-0 seedlings that were pretreated without or with 1 μM GNS for 6 h. Seedlings were treated with 5 μM GA$_3$ at indicated times. **c** AlphaScreen analysis of biotin–GID1A–FLAG–GARU interaction in the presence (+) or absence (−) of 1 μM GA$_3$ and V5-GAI. Data are means (±s.d.) of three independent experiments ($n = 3$). **d** AlphaScreen analysis of biotin–GID1A–FLAG–GAI interaction in the presence (+) or absence (−) of 1 μM GA$_3$ and V5-GARU. Data are means (±s.d.) of three independent experiments ($n = 3$). **e** In vitro ubiquitination assay of GID1A. GID1A-V5 was incubated with FLAG-GARU$^{WT}$ (W) or GARU$^{C190/193S}$ (CS) in the presence of 1 μM GA$_3$, and biotin-DHFR or biotin-RGL2. Ubiquitinated GID1A (Ub-V5-GID1A) and unubiquitinated GID1A (V5-GID1A) were detected by anti-V5 antibody. **f** Immunoblot analysis of GID1A-AGIA. GID1A-AGIA was transfected into protoplasts. After incubation with 10 μM GA$_3$ and/or 5 μM GNS for 6 h, GID1A-AGIA and ubiquitinated (Ub) GID1A-AGIA proteins were analyzed by anti-AGIA antibody

In this study, we found a GARU for ubiquitination of GA receptor, GID1. Interestingly GARU could not ubiquitinate the GID1 protein in the presence of excessive GA and DELLA, although the GARU–GID1 interaction did not change (Figs. 2e, 5c–f, purple dotted line in Fig. 8). From these results, we infer that the amount of GA in the cells influences GARU function. Taken together, these results suggest that GARU plays a role of negative regulator for GA signaling by destabilizing free GID1 protein levels in the nucleus (black lines in Fig. 8). A previous study has indicated that GID1s protein levels but not mRNA levels were increased by after-ripening or imbibition in the seeds[33], suggesting that the GID1 protein level could be regulated at the translational step or by post-translational modification. In this study, we found that the amount of GID1s protein in dry seed is negatively regulated by GARU and its regulation mechanism is disrupted by imbibition of water (Fig. 7a, b, g). Our studies further revealed that GARU negatively regulates the amount of GID1s protein in seeds under salt stress (Fig. 7c–e, g), and GA$_3$ dramatically suppresses it (Fig. 7f). However, under salt stress, germination of Col-0 was not rescued to the same level as garu-1

mutant by GA$_3$ treatment (Fig. 7h). These results suggested that GID1 protein in garu-1 mutant is more stable than Col-0 under salt stress. It may be not possible to stabilize GID1s protein over a long period by the GID1–GA–DELLA complex formation under salt stress because DELLA protein is rapidly degraded. Taken together, GARU acts as a negative regulator and suppresses seed germination under salt stress through the degradation of GID1 protein (brown lines in Fig. 8).

Our studies revealed that the C-terminal region of GARU is important for the interaction between GARU and GID1A, and Tyr321 in this region was phosphorylated by TAGK2, resulting in the Tyr phosphorylation of GARU suppressed the interaction between GARU and GID1A (Fig. 4). In addition, ubiquitin-dependent degradation of GID1A was decreased by TAGK2-dependent phosphorylation of GARU on the Tyr321 residue in protoplasts. These results suggest that TAGK2 plays a role of negative regulator for GARU. In this study, we further uncovered that TAGK2 is a target of GNS in plant protein kinases (Figs. 3, 4e and Supplementary Fig. 4). Furthermore, we observed that TAGK2$^{WT}$-OX1 plants were changed the response to GNS and

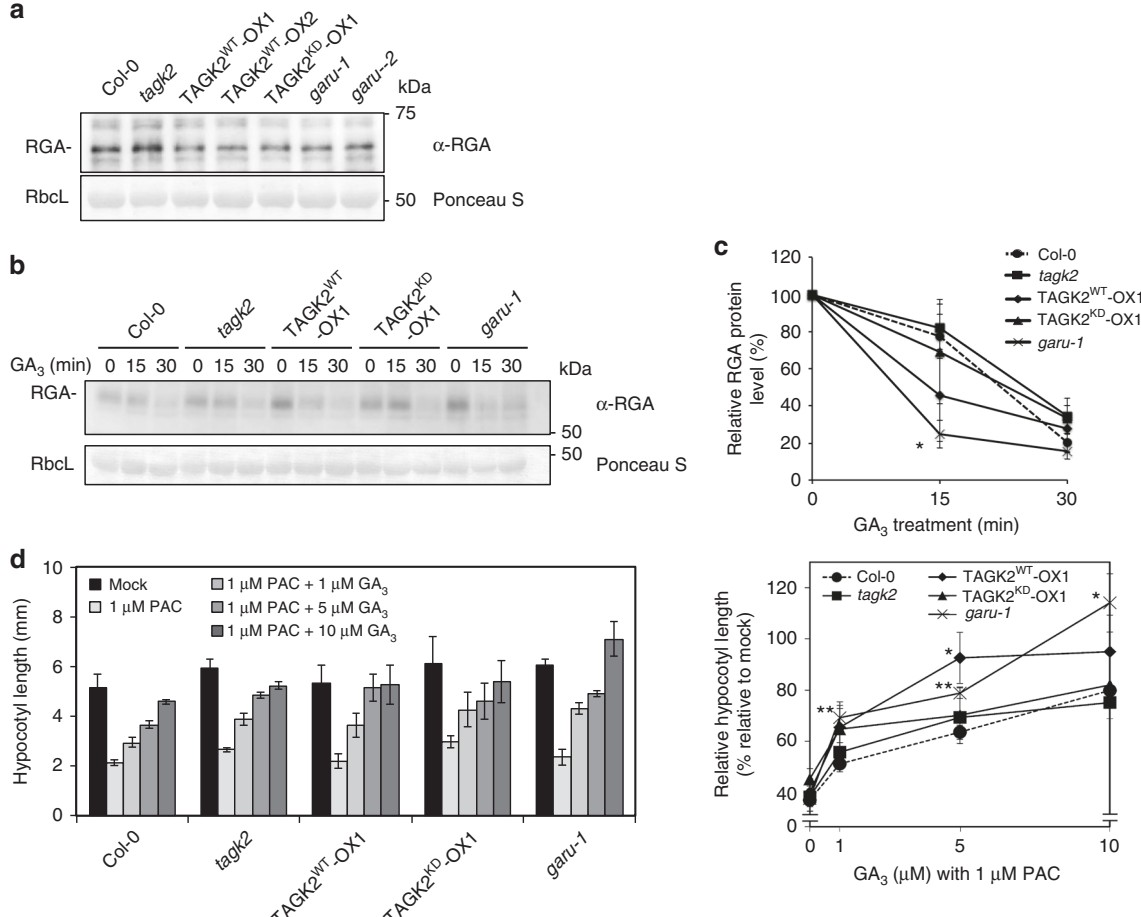

**Fig. 6** GARU and TAGK2 mediate sensitivity of GA response in the plant. **a** Immunoblot analysis of GID1A/C in Col-0, *tagk2*, TAGK2$^{WT}$-OX1, TAGK2$^{WT}$-OX2, TAGK2$^{KD}$-OX1, and *garu-1*. **b, c** GA-induced RGA degradation assay in Col-0, *tagk2*, TAGK2$^{WT}$-OX1, TAGK2$^{KD}$-OX1, and *garu-1* seedlings. Seedlings were treated with 5 µM GA$_3$ at indicated times, and RGA proteins was detected by immunoblot analysis with anti-RGA antibody (**b**). Signal value of RGA proteins and RbcL were measured by ImageJ software. RGA protein levels were normalized to RbcL, and the value of the starting point was set to 100% (**c**). Data are means (±s.d.) of three independent experiments (*n* = 3). Statistically significant changes compared with in the starting point are indicated (*$P <$ 0.05, as determined using two-tailed Student's *t*-test). **d** Hypocotyl elongation assay of etiolated seedlings of Col-0, *tagk2*, TAGK2$^{WT}$-OX1, TAGK2$^{KD}$-OX1, and *garu-1* in the presence or absence of GA$_3$ and PAC. Hypocotyl length was measured using ImageJ software (left). Relative hypocotyl length was calculated as the ratio of the mock treatment (right). Error bars indicate ±s.d. of three replicates, *n* = 24. Statistically significant changes compared with Col-0 are indicated (*$P <$ 0.05, **$P <$ 0.01, as determined using two-tailed Student's *t*-test)

GA$_3$. In TAGK2$^{WT}$-OX1 plants, GNS-induced destabilization of GID1s was not observed (Fig. 4j). Similar to *garu* mutant, TAGK2$^{WT}$-OX1 showed a high sensitivity to exogenous GA$_3$ (Fig. 6). These results suggest that TAGK2-dependent Tyr phosphorylation acts as an inhibitory switch by preventing GARU–GID1 complex formation and consequently promotes the stabilization of GID1s protein (red lines in Fig. 8). This finding indicates that the biochemical and physiological functions of Tyr trans-phosphorylation are evident in plants. Since the TAGK2-phosphorylated Tyr residue is conserved in GARU orthologs of dicot and monocot plants, the regulation of GA receptor by Tyr phosphorylation may be widely used in many plants. However, contrary to our expectation, *tagk2* mutant plants showed that GID1 proteins were decreased by treatment with GNS and sensitivity to GA$_3$ was similar to that in WT plants (Figs. 4j, 6). These results suggest that the GARU is phosphorylated by TAGK2 as well as other TAGKs[22] or that specific, but unknown conditions are required for activation of TAGK2.

In plants, GNS treatment dramatically inhibits DELLA degradation[11, 12]. Accumulation of DELLA may be caused by the following factors: decrease of GA level[34], decrease of GA signaling factors, such as GID1[35] or F-box protein SLY1[34], avoidance from decomposition by post-translational modification of DELLA[9, 36]. Furthermore, since DELLA accumulation by GNS treatment was observed after 6 h (Fig. 1b), it may be caused by regulation of protein function, but not by gene expression regulation. However, how GNS works in GA signaling and what its target protein is remains unclear. In this study, as a one of the mechanism of GNS-induced DELLA accumulation, we found that GNS strongly inhibits the activity of TAGK family (Figs. 3, 4e) and thereby induced DELLA stabilization in three steps: (1) GARU is activated by GNS via inactivation of TAGK2, (2) subsequently GARU induces ubiquitin-dependent degradation of GID1s, and (3) finally DELLA is stabilized by reduction of GID1s protein level in the cell (blue lines in Fig. 8). Although GNS inhibited the activity of TAGK2 at very low concentrations (Fig. 3), hypocotyl elongation and accumulation of DELLA were observed at higher concentrations than in vitro analysis (Fig. 1a, b). Differences in sensitivity to GNS by in vivo and in vitro experiments are thought to be due to its membrane permeability and intracellular stability. In particular, the enzyme of GNS metabolizing may be actively functioning in *Arabidopsis* because intracellular contents of GNS

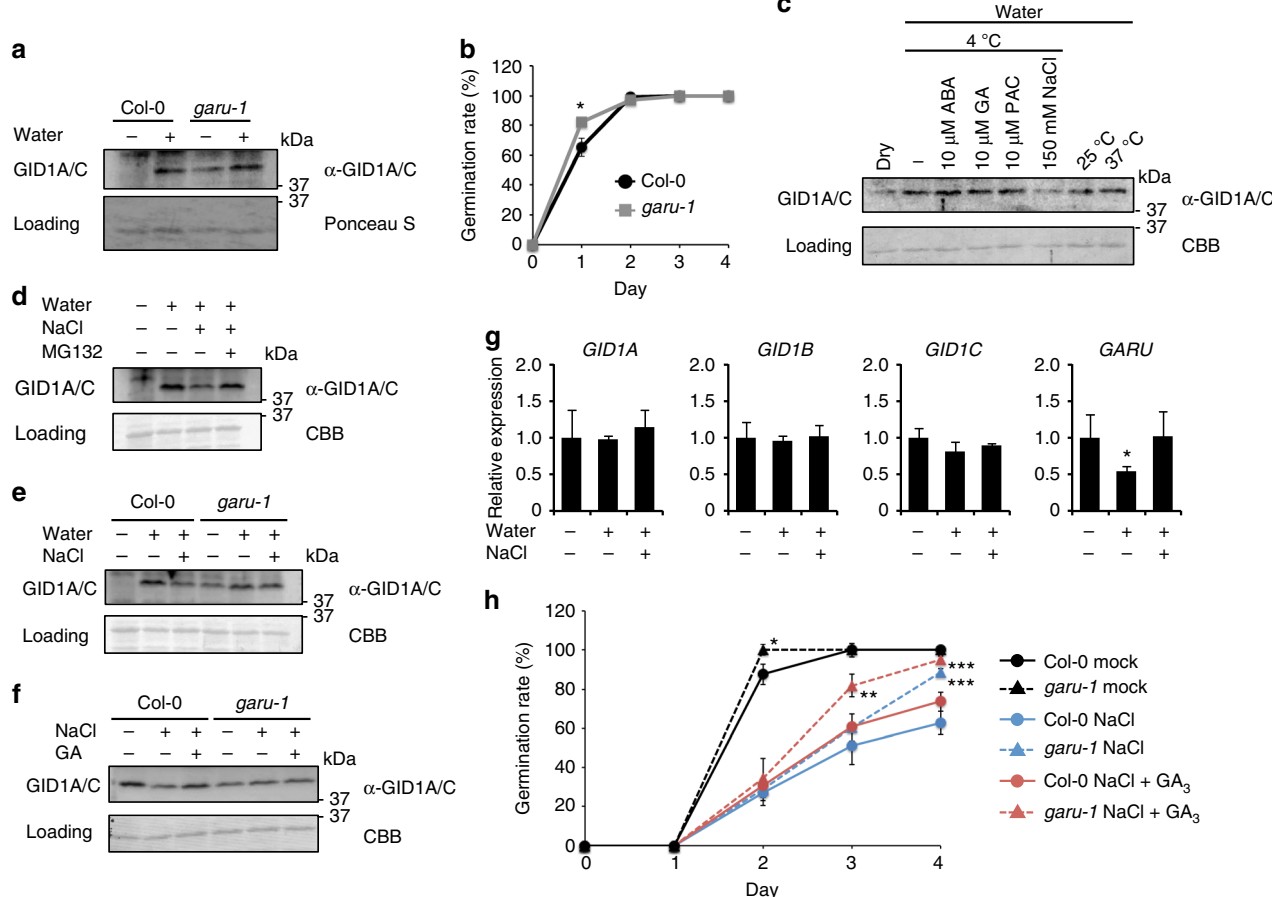

**Fig. 7** GARU mediate sensitivity of GA response in the seed under the salt stress. **a** Immunoblot analysis of GID1A/C in Col-0 and *garu-1* seeds that were treated with (+) or without (−) of water for 12 h at 4 °C. **b** Germination efficiency of Col-0 and *garu-1* seeds in water. **c** Immunoblot analysis of GID1A/C in the Col-0 seeds. Seeds were incubated in the indicated conditions for 12 h. **d**–**f** Salt-mediated instability analysis of GID1A/C in seeds by immunoblot analysis. After incubation of Col-0 seeds with (+) or without (−) of water and 150 mM NaCl solution for 12 h at 4 °C, seeds were incubated with 40 μM MG132 for 6 h (**d**). The Col-0 and *garu-1* seeds were incubated with (+) or without (−) of water and 150 mM NaCl solution for 12 h at 4 °C (**e**), and then seeds were treated with 10 μM $GA_3$ for 6 h (**f**). **g** RT-qPCR analysis of *GID1A*, *GID1B*, *GID1C*, and *GARU* genes in Col-0 seeds. Seeds was treated with (+) or without (−) of water or 150 mM NaCl solution for 12 h at 4 °C. Transcript levels were normalized to the level of UBC21, and relative transcript level in untreated seeds as one. Experiment had three biological repeats and the average value is shown with s.e.m. (*n* = 9). Statistically significant changes compared with dry seeds are indicated (*$P < 0.05$, two-tailed Student's *t*-test). **h** Germination efficiency of Col-0 and *garu-1* seeds under the salt stress condition. After stratification in 150 mM NaCl solution or water for 2 days at 4 °C, seeds were germinated on solid medium containing with or without (mock) of 150 mM NaCl or 150 mM and 10 μM $GA_3$. The percentage seed germination (defined by radicle protrusion) was measured every 24 h after stratification. Data are means (±s.d.) of three independent experiments of 50 seeds each are presented (*n* = 3 in Fig. 7b and *n* = 6 in Fig. 7h), and statistically significant changes compared with Col-0 under the same conditions are indicated (*$P < 0.05$, $0.01 < $**$P < 0.05$, ***$P < 0.01$, two-tailed Student's *t*-test) (**b**, **h**)

metabolites are higher in comparison with GNS[16]. The function of GNS is poorly understood in plant. Thus, there may be other targets of GNS.

There is a direct correlation between the GA signaling and endogenous GA level in plant. Accumulation of GA activates the GA signal by promoting the GID1-depnedent degradation of DELLA. However, DELLA has a function to induce the expression of GA biosynthesis genes and GID1 genes, and excessive accumulation of GA causes decrease of endogenous GA and GID1 protein[37, 38]. These mechanisms are one of the important feedback systems controlling the GA signal. Besides the mechanism that controls GA synthesis/metabolism and gene expression, previous studies have been suggested that plants have multiple mechanisms to promote or suppress GA signaling by regulating the function of DELLA, such as serine/threonine phosphorylation[9, 10]. In this study, our biochemical and genetic data strongly support the notion that GA signaling is negatively regulated by GARU-dependent GID1 ubiquitination and positively regulated by TAGK2 through Tyr phosphorylation and inactivation of GARU. Furthermore, our study suggested that GARU negatively regulates GA sensitivity via degradation of GID1 in the dry seeds and under the salt stress. However, deletion of *GARU* gene and overexpression of *TAGK2* gene did not show an excessive effect on growth and germination. Although stabilization of GID1 is expected to cause an increase in GA signal, destabilization of DELLA causes decrease of endogenous GA and GID1 protein. Furthermore, our study suggested that GID1 is degraded by GARU, but GA signaling through the GID1–GA–DELLA complex formation is more predominant than GID1 degradation (pink lines in Fig. 8). Thus, in order to maintain homeostasis of GA signal, the balance between GA levels and signal transduction is most likely controlled by multiple mechanisms such as gene expression, GA biosynthesis/metabolism, and post-translational modifications[9, 10, 36–39].

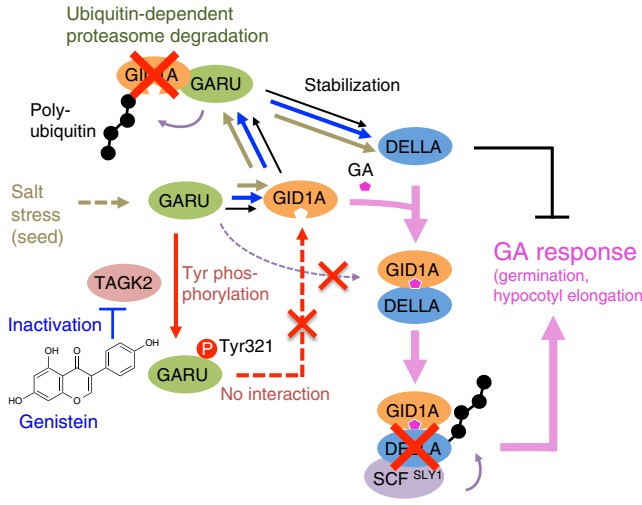

**Fig. 8** A proposed model for control of GID1A protein stability by GARU-dependent ubiquitination and TAGK2-dependent Tyr phosphorylation. GARU induces ubiquitin-dependent proteasome degradation of GID1A (black lines). Destabilization of GID1A causes the stabilization of the DELLA, and GA signal is suppressed. However, GARU cannot ubiquitinate GA–GID1A–DELLA complex (purple dotted line), indicating that GA-dependent complex formation is more predominant than GID1A degradation by GARU. GARU–GID1A interaction is regulated by TAGK2-dependent Tyr phosphorylation of Tyr321 residue on GARU (red lines). In addition, GNS strongly inhibits the activity of TAGK2 and thereby induces stabilization of DELLA by GARU-dependent GID1A degradation (blue lines). In the seeds, GARU mediates the suppression of seed germination under the salt stress through the degradation of GID1 protein (brown lines)

GARU and TAGK2 are also probably one of the regulation mechanisms, and these play a role in regulating GID1 protein level.

## Methods

**Plant materials**. The ecotype Columbia (Col-0) was the parent strain for all mutants and transgenic lines in this study. Mutant of *tagk2* (Salk_090938C) was described previously[22]. *garu-1* (Salk_037121C), *garu-2* (Salk_127503C), and *qDELLA* (gai-t6 rga-t2 rgl1-1 rgl3-1 rgl2-1/SGT625-5 quintuple *della* mutant, CS16298) were obtained from ABRC (https://abrc.osu.edu/). Transgenic seedlings of *TAGK2*[WT]-OX1, *TAGK2*[WT]-OX2, and *TAGK2*[KD]-OX1 were obtained through agrobacterium-mediated transformation of *Arabidopsis* using the floral dip method[40] in WT Col-0. Transgenic plants were screened with 50 µg/mL Kanamycin on MS medium. Homozygous lines were identified by comparison with the parental phenotype and through PCR-based genotyping. All seeds were kept dry at room temperature for 1 year in the dark. For growth assay and immunoblot analysis, WT Col-0, Landsberg erecta (Ler), transgenic plants and mutants seeds were surface sterilized, rinsed with sterile water, and stratified at 4 °C for at least 3 days. Then, seeds were germinated on germination medium (GM) (without sucrose) containing 0.8% agar in a growth cabinet at 22 °C under a 16-h light/8-h dark photoperiod.

**Chemicals**. Gibberellin A3 (GA3) (Tokyo Chemical Industry), PAC (Wako Pure Chemical), GNS (Sigma-Aldrich), daidzein (Sigma-Aldrich), genistin (Sigma-Aldrich), MG132 (PEPTIDE INSTITUTE), and CHX (Calbiochem) were prepared as stock solutions by DMSO. All compounds were added to the culture medium or in vitro reaction assay buffer after they diluted it in DMSO as needed. The final concentration of DMSO was 1% or less.

**Plasmids construction**. The genes of GID1A, GID1B, GID1C, GARU, RGL2, GAI, PYL1, PYR1, and UBQ10 were amplified by PCR from the RIKEN *Arabidopsis* full-length (RAFL) cDNA library. GID1A[K0] mutant was synthesized by GeneArt Gene Synthesis (Thermo Fisher Scientific). The DNA fragments of the open-reading frame (ORF) were cloned in pDONR221 vector using the gateway cloning system (Thermo Fisher Scientific). To construct expression plasmid of linked three HA-UBQ10 genes, ORF region was amplified by PCR with PCR primer pairs (S1-HA-

UBQ10 and R-UBQ-SpeI), (F-UBQ-SpeI and R-UBQ-AatII), (F-UBQ-AatII, T1-UBQ10) using UBQ10 in pDONR221 as a template. Amplified three DNA fragments were digested by *Spe*I and *Aat*II, and they were ligased by T4 DNA ligase. After purification, 3 × (HA-UBQ10) DNA fragment was subcloned into pCR2.1-TOPO vector by TA-cloning method (Thermo Fisher Scientific). The 3 × (HA-UBQ10) DNA fragment was amplified by PCR, and were cloned into pDONR221 vector. After confirming sequence, we generated these expression vectors by LR clonase recombination with pEU vectors for cell-free protein synthesis or transient expression vectors. For mutation analysis, mutagenesis was carried out using a PrimeSTAR Mutagenesis Basal kit (Takara Bio), according to the manufacturer's instructions. The mutated genes were sequenced using an ABI PRISM 310 DNA sequencer (Applied-Biosystems). All primer sequences used in this study are listed in Supplementary Table 2.

For Agrobacterium-mediated transformation, DNA fragment of C-terminal HA-tagged WT or KD form TAGK2[22] was cloned into the binary destination vector pBCR-79[41] by LR clonase recombination, and finally pBCR-35S:::TAGK2[WT]-HA or pBCR-35S:::TAGK2[KD]-HA was generated. Then, these were electroporated into *Agrobacterium tumefaciens* GV3101 strain.

All primer sequences and plasmids used in this study are listed in Supplementary Tables 2 and 3.

**Immunoblot analysis**. The protein samples were separated by SDS-PAGE and blotted onto polyvinylidene difluoride membranes (Millipore). Epitope-tagged proteins were detected using HRP-conjugated antibodies (anti-AGIA-HRP antibody[25] (1:10,000 dilution), anti-V5-HRP antibody (1:10,000 dilution; Thermo Fisher Scientific)), anti-HA-HRP antibody (1:5000 dilution; Roche Applied Science), and anti-FLAG M2-HRP (1:5000 dilution; Sigma-Aldrich) and anti-His (1:1000 dilution; GE Healthcare) primary antibody. Biotinylated proteins were detected using HRP-conjugated anti-biotin antibody (1:10,000 dilution; Sigma-Aldrich) or Alexa Fluor 647-conjugated Streptavidin (1:5000 dilution; Thermo Fisher Scientific). Tyr phosphorylation or ubiquitin was detected using HRP-conjugated anti-phospho-tyrosine antibody (4G10) (1:5000 dilution; Millipore) or anti-ubiquitin (1:1000 dilution; Santa Cruz Biotechnology) primary antibody. RGA or α-tubulin was detected using anti-RGA (1:1000 dilution; Agrisera) or anti-α-tubulin primary antibody (1:10,000 dilution; Abcam). GID1A and GID1C were detected using anti-GID1A primary antibody (1:1000 dilution; GenScript) that was raised using synthetic peptides of the N-terminal (MAASDEVNLIESRTC) of GID1A as antigens. The anti-GID1A antibody could react with *Arabidopsis* GID1A and GID1C, but not with GID1B (Supplementary Fig. 10). Chemiluminescent detection was performed using Immobilon Western Chemiluminescent HRP Substrate (Millipore) using the secondary antibody HRP-conjugated Sheep Anti-Mouse IgG ECL Antibody (1:10,000 dilution; GE Healthcare) or Donkey Anti-Rabbit IgG ECL Antibody (1:10,000 dilution; GE Healthcare). All antibodies used for immunoblotting assay were listed in Supplementary Table 4. All uncropped blot images are provided in Supplementary Fig. 11. In Fig. 7, images were adjusted for brightness and contrast with the ImageJ software because the protein expression level is low.

**Growth conditions and chemical treatments**. For hypocotyl elongation assay and root elongation assay, we used 5-days-old seedlings after germination on GM plates. Seedlings were transferred on half-strength Murashige and Skoog agar plates (1% sucrose) supplemented with PAC, GA3, GNS or daidzein, and then incubated in a growth cabinet. For hypocotyl elongation assay, the plates were wrapped in aluminum foil before incubation. Seedlings were photographed, and the ImageJ software was used to measure the seedlings hypocotyl length and root length. For germination assay, 50 seeds were imbibed in 400 µl of distilled water in a 24-well plate. After stratification at 4 °C for 4 days, seeds were incubated with water in a growth cabinet at 22 °C under a 16-h light/8-h dark photoperiod. For germination assay under salt stress, 50 seeds were surface sterilized, rinsed with 150 mM NaCl solution or water, and stratified in 150 mM NaCl solution or water at 4 °C for 2 days. Then, seeds were germinated on solid GM (-Suc) with or without of 150 mM NaCl, or 150 mM NaCl and 10 µM GA3. The percentage seed germination (defined by radicle protrusion) was measured every 24 h. The number of quantified biological repeats and sampling sizes (*n*) is indicated for each graph in figure legends.

**Protein stability and destabilization assays**. For RGA and GID1A/C stability assay by GNS treatment, 5-days-old seedlings after germination were transferred in 1 mL GM (-Suc) liquid medium supplemented with 0.1 µM, 1 µM GNS, or mock (0.1% DMSO) in the presence or absence of 300 µM CHX, and then incubated at 22 °C under light for 6 h. For RGA destabilization assay by GA3 treatment, seedlings or GNS-pretreated seedlings (as above-mentioned method) were transferred in 1 mL GM (-Suc) liquid medium supplemented with 10 µM GA3 (0.1% DMSO), and then incubated at 22 °C under light. For GID1A/C stability/destabilization assay in the seeds, after-ripened Col-0 and *garu-1* seeds (kept dry at room temperature for 1 year) were treated with sterile distilled water supplemented with or without of 10 µM ABA, 10 µM GA3, 10 µM PAC, or 150 mM NaCl for 12 h at 4, 25, or 37 °C. When compounds treatment was additionally performed after the incubation, ten micromoral GA3 or 40 µM MG132 was added and the incubation

was continued for 6 h. All samples were taken at indicated intervals, and frozen immediately in liquid nitrogen. The samples were homogenized in the radio-inmunoprecipitation assay (RIPA) buffer (50 mM Tris-HCl pH 8.0, 150 mM NaCl, 1% NP-40, 0.5% sodium deoxycholate, 0.1% SDS) supplemented with 40 μM MG132, phosphatase inhibitor cocktail (PhosSTOP, Roche), and Protease Inhibitor Cocktail (Sigma), and cleared by centrifugation at 12,000 × $g$ for 10 min. Protein content was measured using the BCA assay (Thermo Fisher Scientific) or the DC Protein Assay (Bio-Rad).

RGA and GID1A/C were analyzed by immunoblot analysis with anti-RGA antibody or anti-GID1A primary antibody. Loading proteins were visualized by Ponceau-S staining and Coomassie brilliant blue (CBB) staining as loading control. Signal values of RGA or GID1A/C proteins and RbcL (as loading control) were measured by ImageJ software (http://imagej.nih.gov). RGA and GID1A/C protein levels were normalized to RbcL, and the value of the starting point was set to 100%.

**Cell-free protein synthesis**. In vitro transcription and wheat cell-free protein synthesis were performed using WEPRO1240 expression kit (Cell-Free Sciences). Transcript was made from each of the DNA templates mentioned above using the SP6 RNA polymerase. The translation reaction was performed in the bilayer mode[42] using WEPRO1240 expression kit (Cell-Free Sciences), according to the manufacture's instruction. For biotin labeling, 1 μl of crude biotin ligase (BirA) produced by the wheat cell-free expression system was added to the bottom layer, and 0.5 μM (final concentration) of d-biotin (Nacalai Tesque) was added to both upper and bottom layers, as described previously[43]. The aliquots were used for the expression analysis and functional characterization.

**Protein–protein interaction (PPI) assays**. For GID1A-interacting RING-type E3 ubiquitin ligase screening using AlphaScreen, we used the *Arabidopsis* 204 RING E3 ligases library previously described[26]. In vitro PPI assays were carried out in a total volume of 15 μl containing 1 μl of biotinylated GID1A, and 1 μl of FLAG-RING-type E3 ubiquitin ligase in the AlphaScreen buffer (100 mM Tris-HCl, pH 8.0, 0.1% Tween20, 1 mg/ml BSA) at 25 °C for 1 h in a 384-well Optiplate (PerkinElmer). In accordance with the AlphaScreen IgG (ProteinA) detection kit (PerkinElmer) instruction manual, 10 μl of detection mixture containing AlphaScreen buffer, 5 μg/mL anti-FLAG M2 antibody (Sigma-Aldrich), 0.1 μl of streptavidin-coated donor beads, and 0.1 μl of Protein A-coated acceptor beads were added to each well of the 384-well Optiplate, followed by incubation at 25 °C for 1 h. Luminescence was analyzed using the AlphaScreen detection program. All data represent the average of two independent experiments and the background was controlled using a dihydrofolate reductase (DHFR) from *E. coli*.

To analyze PPI between phospho-GARU and GID1A by AlphaScreen, in vitro phosphorylation reaction was carried out in a total volume of 10 μl containing 5 μl of FLAG-GARU, and 1 μl of TAGK2-His in the kinase buffer (50 mM Tris-HCl, pH 7.6, 100 mM potassium acetate, 10 mM MgCl₂, 1 mM DTT, and 100 μM ATP) at 26 °C for 90 min. After incubation, 1 μl of in vitro kinase assays mixture (containing of 0.5 μl FLAG-GARU) was used for AlphaScreen assay. AlphaScreen assay was carried out as described above. Data were expressed as relative values normalized by the value of the PPI signal between non-phosphorylated WT FLAG-GARU and biotin-GID1A. All data represent the average of three independent experiments. Phosphorylated FLAG-GARU was visualized by immunoblot analysis with anti-phospho Tyr antibody (4G10).

For AlphaScreen-based PPI analysis of GID1A–GARU or GID1A–GAI interaction in the presence of GA₃, we synthesized biotin-GID1A, FLAG-GARU, FLAG-GAI, V5-GARU, and V5-GAI by cell-free system. In vitro PPI assays were carried out in a total volume of 15 μl containing AlphaScreen buffer, 0.5 μl of biotin-GID1A, 0.5 μl of FLAG-GARU or FLAG-GAI, 0.5 μl of V5-GARU or V5-GAI, and 1 μM GA₃ (final 1% DMSO) at 25 °C for 1 h in a 384-well Optiplate. After incubation, the detection was carried out by operation as described above. All data represent the average of three independent experiments.

To in vitro pull-down assay, in vitro phosphorylation reaction were carried out in a total volume of 50 μl containing kinase buffer, 20 μl of HA-GARU, and 5 μl of FLAG-TAGK2 or FLAG-TAGK3 at 26 °C for 60 min. After incubation, 20 μl of GID1A-V5 or V5-PYR1 was mixed, and then was incubated at 26 °C for 60 min. Then, 20 μl of Protein G Sepharose 4 Fast Flow (GE Healthcare), 0.5 μl of anti-V5 antibody, and 200 μl of wash buffer (50 mM Tris-HCl, pH 8.0, 150 mM NaCl, 1% Triton X-100) were mixed. After incubation for 1 h, the beads were washed five times with wash buffer. The 50 μl of 3 × SDS sample buffer was added after complete removal of the wash buffer, and boiled for 5 min. Then, HA-GARU and GID1A-V5 or V5-PYR1 were analyzed by immunoblot analysis with anti-HA-HRP antibody and anti-V5-HRP antibody.

**In vitro ubiquitination assays**. The wheat germ-based ubiquitination assays using unpurified recombinant proteins were carried out as previously described with slight modifications[26, 27]. The assays were performed in a 20 μl reaction mixture containing 10 μl of GID1A (GID1A-V5, biotin-GID1A, and GID1A-AGIA protein were used), 2 μl of FLAG-GARU, at1g47579 or GARU mutants in the buffer (80 mM Tris-HCl, pH 7.5, 12 mM ATP, 1.2 μM UbcH5b (Enzo Life Sciences), 400 nM recombinant ubiquitin (Boston Biochem)) at 26 °C for 0.5–3 h. The

reactions were terminated by the addition of 10 μl of 3 × SDS sample buffer and boiling for 5 min.

For denaturing in vitro ubiquitination assay, ubiquitination reaction was performed in a 20 μl reaction mixture containing 10 μl of biotin-GID1A, GID1B or GID1C, 2 μl of FLAG-GARU or at1g47579 in the buffer (80 mM Tris-HCl, pH 7.5, 12 mM ATP, 1.2 μM UbcH5b (Enzo Life Sciences), 400 nM recombinant HA-ubiquitin (Boston Biochem)) at 26 °C for 2 h. After incubation, reaction mixtures were boiled in 2% SDS for 5 min, and then biotin-GID1s were bond with Dynabeads M-280 streptavidin (Thermo Fisher Scientific). The beads were washed three times with wash buffer (50 mM Tris-HCl, pH 8.0, 150 mM NaCl, 2% SDS). The beads were finally boiled in 20 μl of 2 × SDS sample buffer for 5 min, and supernatants were collected for analysis.

For ubiquitination assays using purified recombinant proteins, GID1A-His and FLAG-GARU were purified using Ni-sepharose High Performance beads (GE Healthcare) or anti-FLAG M2 affinity agarose gel (Sigma-Aldrich). The purified proteins were quantified by CBB stain after were separated by SDS-PAGE (BSA as a protein standard). In viro ubiquitination assay was carried out in a 20 μl reaction mixture containing 200 ng GID1-His and 10 ng FLAG-GARU in the buffer (80 mM Tris-HCl, pH 7.5, 12 mM ATP, 200 nM rabbit UBE1 (Boston Biochem), 1.2 μM UbcH5b (Enzo Life Sciences), 500 nM recombinant ubiquitin (Boston Biochem)) at 26 °C for 2 h. The reactions were terminated by the addition of 10 μl of 3 × SDS sample buffer and boiling for 5 min.

All samples were analyzed by immunoblot analysis with anti-V5-HRP antibody, anti-biotin-HRP antibody, anti-AGIA-HRP antibody[25], anti-His antibody, anti-ubiquitin antibody, anti-HA antibody, or anti-biotin antibody.

**In vitro kinase assays and inhibitor assays**. For in vitro kinase assays, FLAG-TAGK2, FLAG-TAGK3, and FLAG-GARU were purified using anti-FLAG M2 affinity agarose gel (Sigma-Aldrich). Then, in vitro kinase assays were carried out in a total volume of 20 μl containing 100 ng FLAG-GARU, 20 ng FLAG-TAGK2 or FLAG-TAGK3 in the kinase buffer at 26 °C for 60 min. The reactions were terminated by the addition of 10 μl of 3 × SDS sample buffer and boiling for 5 min.

In vitro inhibitor assays were carried out in a total volume of 50 μl containing kinase buffer, purified 50 nM biotin-protein kinase, and 1 μM biotin-ERF13 supplemented with inhibitor or 1% DMSO as control at 26 °C for 30 min. The reactions were terminated by the addition of 25 μl of 3 × SDS sample buffer and boiling for 5 min.

Tyrosine phosphorylation was detected by immunoblot analysis with anti-phospho-Tyr antibody, and loading proteins were detected by Streptavidin Alexa Fluor 647 or anti-FLAG M2 antibody HPR. The signal value was measured by ImageJ software. Concentration-inhibition curves for GNS were quantified from at least three independent in vitro kinase assays. IC₅₀ values were determined by curve fitting.

**In cell ubiquitination and phosphorylation assays**. Isolation of *Arabidopsis* WT Col-0 protoplasts and polyethylene glycol-mediated DNA transfection was performed by transient expression in *Arabidopsis* mesophyll protoplast method[44].

For GID1A-AGIA stability assay by chemical treatment, 10 μg plasmid DNA of GID1A-AGIA was transfected into 2 × 10⁵ protoplasts. After overnight incubation in the dark, protoplasts were treated with 10 μM GA₃, 5 μM GNS, 20 μM MG132, or mock (0.1% DMSO) for 6 h. After protoplasts were harvested, they were mixed with 2 × sample buffer and boiled for immunoblot analysis.

For GARU-dependent GID1 destabilization assay, 10 μg plasmid DNA of V5-GARU or V5-GARU$^{CS}$ and 20 μg plasmid DNA of GID1A-AGIA (GID1A-AGIA was not transfected for the analysis of endogenous GID1) were transfected into 2 × 10⁶ protoplasts. After overnight incubation in the dark, protoplasts were treated with 20 μM MG132 or mock (0.1% DMSO) for 6 h. Harvested protoplasts were mixed with 2 × sample buffer and boiled for immunoblot analysis.

For *in cell* ubiquitination analysis of GID1A-AGIA, 20 μg plasmid DNA of GID1A-AGIA, 10 μg plasmid DNA of V5-GARU$^{WT}$ or V5-GARU$^{CS}$, and 10 μg plasmid DNA of 3 × (HA-UBQ10) were transfected into 2 × 10⁶ protoplasts. After overnight incubation in the dark, protoplasts were treated with 20 μM MG132 (0.1% DMSO) for 6 h. Then, crude extracts were obtained by homogenizing protoplasts in the RIPA assay buffer supplemented with 40 μM MG132, phosphatase inhibitor cocktail (PhosSTOP, Roche), and Protease Inhibitor Cocktail (Sigma). Immunoprecipitation of GID1A-AGIA was performed by mixing protoplast lysate with 50 μl of anti-AGIA antibody-conjugated sepharose beads[25] by rotation at 4 °C overnight. The beads were washed four times with RIPA buffer containing 1 M NaCl. Then, the beads were finally boiled in 40 μl of 2 × SDS sample buffer for 5 min, and supernatants were collected for analysis.

For *in cell* phosphorylation analysis of FLAG-ERF13, 40 μg plasmid DNA of FLAG-ERF13, and 5 μg plasmid DNA of TAGK2-HA or TAGK3-HA were transfected into 2 × 10⁶ protoplasts. After overnight incubation in the dark, protoplasts were treated with 10 μM GNS or mock (0.1% DMSO) for 6 h. Then, crude extracts were obtained by homogenizing protoplasts in immunoprecipitation buffer (50 mM Tris-HCl, pH 8.0, 150 mM NaCl, 1% Triton X-100) supplemented with phosphatase inhibitor cocktail (PhosSTOP, Roche) and Protease Inhibitor Cocktail (Sigma). Immunoprecipitation of FLAG-ERF13 was performed by mixing protoplast lysate with anti-FLAG M2 affinity agarose gel (Sigma-Aldrich) by rotation at 4 °C overnight. The beads were washed four times with

immunoprecipitation buffer, and boiled in 40 μl of 2 × SDS sample buffer for 5 min. Supernatants were collected for immunoblot analysis.

For *in cell* phosphorylation analysis by Phos-tag SDS-PAGE, 20 μg plasmid DNA of V5-GARU$^{CS}$ or AGIA-GARU$^{CS}$ and 5 μg plasmid DNA of TAGK2-3 × FLAG were transfected into 2 × 10$^6$ WT Col-0 or *tagk2* mutant protoplasts. After overnight incubation in the dark, they were treated with 1 μM GNS or mock (0.1% DMSO) for 6 h. After protoplasts were harvested, they were mixed with 2 × sample buffer and boiled. Then, phosphorylated GARU were analyzed by Phos-tag SDS-PAGE (50 μM Phos-tag and 50 μM MnCl$_2$) and immunoblot analysis.

All samples were analyzed by immunoblot analysis using anti-V5-HRP antibody, anti-AGIA-HRP antibody, anti-HA-HRP antibody, anti-FLAG M2 antibody HPR, anti-phospho-Tyr antibody, or anti-GID1A antibody. α-tubulin or RbcL was visualized by anti-α-tubulin antibody or Ponceau-S staining as loading control.

**Subcellular localization analysis**. For subcellular localization analysis and BiFC assay, 10 μg plasmid DNA was transfected to 2 × 10$^4$ protoplasts by transient expression in *Arabidopsis* mesophyll protoplast method[44]. After overnight incubation, protoplasts were treated with or without of 10 μM GA$_3$ for 6 h. YFP and GFP fluorescence were observed with a confocal laser-scanning microscope LSM5 PASCAL (Zeiss) and a fluorescence microscope IX-73 (Olympus). Nuclear were visualized by Hoechst33342 (Thermo Fisher Scientific) staining.

**Gene expression analysis**. Gene expression analysis was performed by reverse transcription quantitative real-time PCR (RT-qPCR) analysis. For RT-qPCR analysis of *GID1A* and *GID1C* expression in seedling, total RNA was isolated form 5-days-old seedling using RNeasy Plant mini kit (Qiagen). For RT-qPCR analysis of *GID1A*, *GID1B*, *GID1C*, and *GARU* expression in seeds, Col-0 seeds were treated with or without of sterile distilled water or 150 mM NaCl solution for 12 h at 4 °C. After incubation, seeds were homogenized, and total RNA was extracted using RNeasy Plant mini kit (Qiagen). One hundred nanograms of extracted total RNA were used for RT-qPCR analysis using the KOD SYBR qPCR/RT Set III (TOYOBO), according to the manufacture's instruction. The RT-qPCR analysis was carried out with a real-time LightCycler96 PCR system (Roche) with gene-specific primer pairs. The relative quantification was calculated with the ΔΔCt method and ACTIN2 (at3g18780) or UBC21 (at5g25760) was analyzed as internal control genes. All primer sequences used in this study are listed in Supplementary Table 2.

**Prediction of AtTAGK3-GNS binding poses**. We found nine kinase domain structures interacting with inhibitors sharing the same two aromatic rings that the benzopyrone structure in GNS has, as exemplified by apigenin and quercetin in the Protein Data Bank (http://www.rcsb.org/pdb/). Pairwise sequence identities were high (56–65%) between the AtTAGK3 kinase domain and the nine kinase domains: 3LM5A, 2O3PA, 2O63A, 2O64A, 2O65A[45], 4DGNA, 4DGMA[46], 3BLRA[47], and 2HCKA[48]. The two aromatic rings of all inhibitors are located near the hinge region of ATP-binding pocket in different combinations of kinase domains and inhibitors, but there were two inhibitor orientations: the two aromatic rings are located inside or outside of the ATP-binding pocket. Based on these observations, we hypothesized that GNS is located at the ATP-binding pocket of the AtTAGK3 kinase domain similarly to the inhibitors of the nine kinase domains. To sample the potential binding poses of GNS interacting with the AtTAGK3 kinase domain (Supplementary Fig. 6a, d), we set GNS in the ATP-binding pocket of nine kinase domains to superpose the two aromatic rings of GNS over those of inhibitors. Then, we executed homology modeling with GNS using the nine structures as templates. Homology modeling was done using the Homology Model function of the Molecular Operating Environment (MOE) software package (Chemical Computing Group, MOE, ver. 2012.10, www.chemcomp.com) with the Amber12:EHT force field. We generated 200 independent kinase domain models interacting with GNS for each template. To scrutinize details of those models, we specifically examined the difference between GNS and DIZ: AtTAGK3 was inhibited by GNS, but not by DIZ, although the difference between the two inhibitors was only one OH group at the 5-site of the two aromatic rings. Therefore, we assume that the OH group at the 5-site of GNS (position 1 in Supplementary Fig. 6b) was related to inhibition of AtTAGK3. We selected 245 out of 1800 models of the AtTAGK3 kinase domain interacting with the OH group at the 5-site of GNS as prediction models. To analyze and classify binding poses of the 245 models, we made a binary fingerprint for the 25 observed binding residues (K147, V149, G150, F154, T157, K175, 194E, L207, K209, V223, M224, E225, L226, C227, D228, G229, G230, E231, D234, L278, T280, S281, R283, I291, and D292) interacting with GNS in and around the ATP-binding pocket based on both the 11 observed binding positions of GNS (Supplementary Fig. 6b) and the 5 interaction patterns (side chain hydrogen bond donor/acceptor, backbone hydrogen bond donor/acceptor, and arene attraction). We observed 91 unique fingerprints and then performed hierarchical clustering of the binding poses based on those fingerprints using Ward's method with Euclidean distance as implemented using R software [49]. Consequently, we obtained 22 clusters as representatives that roughly reflect the classification based on residues that interact with the OH group at the 5-site of GNS (Supplementary Fig. 6c, d). Among these modes, we found that the OH group frequently interacted with M224 or C227 in the hinge region of the kinase domain,

and thus selected a model with such an interaction and a low free binding energy as a representative model (Supplementary Fig. 6c, d).

**Multiple sequence alignment**. Identification of *A. thaliana*, *G. max*, *Z. maize*, *S. lycopericum*, *H. vulgare*, *O. sativa* genes homologous to *Arabidopsis* GARU gene was done using the BLAST program in UniProt (http://www.uniprot.org/blast/). Then, we selected one or two homologs genes in the species that showing highest alignment scores, and the sequence alignments were done using ClustalW using the tools of the DDBJ program (http://clustalw.ddbj.nig.ac.jp/).

**Statistics and reproducibility of experiments**. The quantitative values obtained in the figures were analyzed in Excel spreadsheets with the embedded basic statistical functions. Sample size was not predetermined using statistical methods, but took into account the variability of the traits measured, assessed by the s.d. No samples were excluded from the analyses. The experiments were not randomized, and investigators were not blinded to allocation during experiments and outcome assessment. Data sets with normal distributions were analyzed using two-sided, unpaired Student's *t*-tests and presented as the means ± s.d. (Fig. 7g presented as means ± s.e.m.). The results for statistical significance tests are included in the legend of each figure, and *n* values represent the number of independent experiments performed or the number of seedlings. All in vivo and in cell assays were repeated at least three times, and all in vitro assay were repeated at least two times.

**Data availability**. The authors declare that all data supporting the findings of this study are available in the manuscript and its supplementary files or are available from the corresponding author upon reasonable request.

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

## Acknowledgements

We thank the Applied Protein Research Laboratory of Ehime University, Mr. Naoyuki Asakawa, Ms. Maho Daito, and Mr. Takumi Miyamoto for technical assistance. This work was mainly supported by the Platform for Drug Discovery, Informatics, and Structural Life Science (T.S. and K.T.) from Japan Agency for Medical Research and Development (AMED), and a Grant-in-Aid for Scientific Research on Innovative Areas (JP25117719 and JP16H06579 for T.S., JP16H01471 for G.A.) from Japan Society for the Promotion of Science (JSPS). This work was also partially supported by JSPS KAKENHI (JP16H04729 and JP25290077 for T.S., JP16K07407 for G.A., and JP16K18570 for K.N.).

## Author contributions

T.S. and K.N. conceived the study, designed the experiments, and wrote the manuscript. K.N., A.R. and G.A. performed experiments; K.I. and K.T. did the protein modeling; K.N., K.S. and T.S. analyzed the data.

## Additional information

**Competing interests:** The authors declare no competing financial interests.

