## [Peer Review File · Nature Communications]

Reviewers' comments:

Reviewer #1 (Remarks to the Author):

In this manuscript, the authors identified GARU to be the E3 ligase of GA receptor GID1, and TAGK2 to be the kinase that phosphorylating GARU at the Tyrosine residues. TAGK2, a target of GNS, can inhibit GARU-GID1A interaction by phosphorylating GARU. So, GNS treatment inhibits TAGK2 kinase activity and releases its inhibition effect on GARU, and then released GARU proteins promote GID1 ubiquitination and degradation. Thus, DELLA proteins are accumulated and GA response is repressed. The manuscript identified two key factors in GID1 stability regulation, which is important and interesting.

Generally, this is a nice story. Here are the comments.

1. About how GA regulates GID1 protein levels:

(a) It is interesting that GA treatment inhibits GID1 ubiquitination, and thus increases GID1 protein level (Figure 1d). An *in vivo* assay in *Arabidopsis* should be added to confirm this. The authors can also check whether GA induced GID1 accumulation is dependent on the level of E3 GRAU. For example, authors may treat Col and *garu* mutant with GA and PAC, and then check GID1 protein levels.

(b) Figure 5c showed that GA has no effect on GID1-GARU interaction. So, it is possible that GA induced GID1-DELLA interaction block the key lysine residues in the GID1. Can this be tested and discussed?

2. It will be better to confirm the shifted bands in figure 1g are "Ub-GARU" and "Ub-GID1" using anti-Ub antibody. Does the result in figure 1g mean that GARU E3 ligase can be self-ubiquitinated?

3. How important is GRAU in plant development? In *garu* mutants growing under mock or normal conditions, GID1 and RGA proteins didn't change and no obvious phenotype was observed (Figure 2e, 6a, 6c-d). GRAU-OX line may be included to check the GID1 protein levels and plant phenotype.

4. In Figure 3, ERF13 is not related to the major pathway studied, which only provides clue for identifying TAGK2 as the kinase of GARU. TAGK3 showed phosphorylation activity toward ERF13 (figure 3e), while TAGK2, instead of TAGK3, showed phosphorylation activity toward GARU (figure 4b), which are different. So, I suggest the authors to move figure 3 into supplemental materials. Additionally, the experiments in figure 3g should be done in the same gel to prove that M224A and C227A mutations actually repress TAGK3 kinase activity.

5. Figure 4h, why do non-phosphorylation form of Y321F and the two phosphorylation-mimic forms of Y321D and Y321E all partially rescue the GID1A expression? They should show opposite effects, right?

6. Figure 4j, in TAGK2WT-OX1 and *garu-1* without GNS treatment, the GID1A/C level should increase comparing to Col-0. Why did GID1 protein levels in this plants decrease comparing Col-0? Why is TAGK2KD-OX1 still regulated by GNS?

7. Figure 6a, why RGA didn't increase in *tagk2* mutant? Figure 6b show this pattern, but the samples are not in the same gel.

8. Is it possible to do some genetic relationship analysis of GID1, GARU and TAGK2?

Minor :

1. The abbreviation in figures should be well illustrated in figure legends. For example, in Figure 3, "pY" in ERF13 (pY) should be explained.

2. In figure 4j and figure 6b, error bars should be labeled in the quantitative statistics results.

3. There are several spelling errors in the manuscripts, for example, "RAG" in line 515, 516 and 518 should be RGA, "constrictions" in 533 should be constructions.

4. The authors should go through the MS to make sure that all the descriptions are clear. For example, in line 315, "To observe the stabilization of GID1A on Tyr321 phosphorylation of GARU", may be

changed to "To observe the stabilization of GID1A under the control of Tyr321 phosphorylation of GARU".

Reviewer #2 (Remarks to the Author):

This manuscript describes the novel finding that the gibberellin (GA) receptor, GID1, is subject to ubiquitination-mediated degradation. The authors used an AlphaScreen-based protein-protein interaction screen to identify the RING E3 ubiquitin ligase (GARU) responsible for GID1 ubiquitination in Arabidopsis and showed that its interaction with GID1 was attenuated by phosphorylation. Furthermore, they identified the tyrosine kinase involved in this phosphorylation, named TAGK2, and demonstrated that its activity is inhibited by genistein, a known kinase inhibitor. Gibberellin promotes plant growth by initiating the degradation of the growth-repressing DELLA proteins in association with GID1. It has previously been shown that genistein stabilised DELLAs, but no evidence could be obtained to suggest that DELLA stability was influenced by its phosphorylation. This manuscript provides an explanation for this observation: by inhibiting the kinase and activating the E3 ligase, genistein destabilises GID1 and thereby stabilises DELLA. This hypothesis is well supported by a series of elegant experiments employing an impressive array of resources. Indeed the Methods section is extremely long. Could some of this material be provided as on-line only?

The authors might comment on the physiological significance of this mechanism, which would appear to have a fairly small influence, at least under normal conditions. Despite IC₅₀ values for genistein against TAGK in the low nM range in vitro, 10 µM was used to inhibit Arabidopsis hypocotyl elongation. While 1 µM genistein treatment accumulated DELLA in seedlings, this concentration reduced GID1A/C content only in the presence of cycloheximide. Loss of GARU or overexpression of the TAGK2 kinase had little effect on hypocotyl length, except when seedlings were treated with GA. It would appear that GID1 is protected from ubiquitination by GARU when in association with DELLA, which may explain why GA, which promotes the GID1-DELLA interaction, stabilises GID1 (Fig. 1d). However, GA promotes DELLA degradation and is known to suppress GID1 gene expression through this DELLA destabilising. Thus the mechanism described in the manuscript forms part of the complex homeostasis in GA biosynthesis/signal transduction. These aspects are not clearly explained in the manuscript.

The statistical methodology, which is critical to the work, is not clearly described.

The section on the modelling of the genistein interaction with TAGK2 is something of a diversion and, although of interest, might be omitted if space is limited.

On page 3, line 49: DELLAs do not only act as negative regulators of transcription. In fact, they commonly act as transcriptional co-activators. Sequestration of transcription factors, as described on line 54, is only one mode of action.

Reviewer #3 (Remarks to the Author):

Sawasaki reports an experimental analysis of the molecular connection between GARU, Genistein and GID1. In my opinion there is not enough novelty in the manuscript and it is not clear the importance of the elucidation of this molecular connection.

Furthermore, the computational section of the paper is completely unclear. It is not clear if they applied docking calculations and which is the pose selection criteria. Finally, a molecular dynamic simulation of the ligand-protein complex should be carried out in order to support the suggested ligand pose.

Reviewer #4 (Remarks to the Author):

GA plays the important roles in the regulation of plant growth and development throughout the plant life-cycle, including seed germination, leaf expansion, stem elongation and floral development. It is known that GA promotes plant growth via the GA receptor GID1-mediated degradation of DELLA proteins, and previous studies have shown that Ser/Thr phosphorylation/de-phosphorylation of the GA-DELLA regulatory system is involved in GA-induced DELLA degradation. However, the molecular mechanism of this modification still remains unclear. In this manuscript, the authors found that GARU E3 ligase was required for 26S-dependent degradation of GID1. Loss-of-function garu mutant plants displayed higher levels of GID1, and consequently resulting in the degradation of DELLA proteins. The authors also found that Tyr-kinase TAGK2 could inhibit the interaction between GARU and GID1 by phosphorylation of GARU, and the up-regulation of TAGK2 also exhibited the similar GA-sensitive phenotypes as the Arabidopsis garu mutant. The topic of this manuscript is sufficient general interest for researchers follow trends and important development in understanding GID1-mediated GA responses, and I think the authors should give attention to the following points and suggestions.

1, DELLA proteins function as plant growth repressor in a dosage-dependent manner. The previous studies have shown that GID1-mediated growth responses is not only dependent on the degradation of DELLA proteins, but also dependent upon the activity of DELLA proteins. For example, the up-regulation of GID1 could promotes growth of the sly-10 mutant. In Fig.1, the authors found that GNS treatment inhibited plant growth, and concluded that GNS specifically induces the instability of GID1 proteins and consequently provides stabilization of DELLA proteins. Indeed, the authors should provide more evidences to consummate his argument. It is interested to know more about differences among wild-type plant, sly1-10 mutant and della mutant (lacking five DELLA proteins). Does GNS treatment affect the hypocotyl length of Arabidopsis gid1 gid1b gid1c triple mutant?

2, the authors identified two RING-type E3 ubiquitin ligases, at2g40830 and at1g4750, that interacted strongly with Arabidopsis GID1A. The authors performed in vitro ubiquitination assays and found that all Arabidopsis GID1 proteins (AtGID1A, AtGID1B and AtGID1C) could be ubiquitinated by the gene product of at2g40830, but ubiquitinated AtGID1A proteins were not detected when AtGID1A was incubated with the gene product of at1g4750. The results of this analysis did not conclude that both AtGID1B and AtGID1C could not be ubiquitinated by the gene product of at1g4750. Thus, the authors should performed in vitro ubiquitination assays using the gene product of at1g4750.

3, to confirm GARU-mediated degradation of GID1, the authors should analyzed the accumulation of Arabidopsis GID1 proteins and their ubiquitination using single and double mutants of at2g40830 and at1g4750.

4, in Fig. 2d, the image of YFP-GARU is not clear. It is interested to know the localization of interaction between GID1A and GARU proteins using BiFC assays. Does GA treatment influence their interaction and/or localization of GARU?

5, as described above, the GA-GID1-DELLA regulatory system plays the important roles in the control of plant growth and development, including seed germination, root extension, hypocotyl elongation and flowering time etc. In this manuscript, the authors only mentioned hypocotyl length. How about other growth phenotypes?

Response to reviewers

We have enclosed our revised manuscript entitled “**Tyrosine Phosphorylation of GARU RING E3 Ligase for Gibberellin (GA) receptor Regulates GA Response**” by Nemoto et al. Thank you for your positive comments on our manuscript and your thoughtful critiques. Below, we have addressed point-by-point your constructive critiques, and modified the text accordingly.

Reviewer #1 (Remarks to the Author):

In this manuscript, the authors identified GARU to be the E3 ligase of GA receptor GID1, and TAGK2 to be the kinase that phosphorylating GARU at the Tyrosine residues. TAGK2, a target of GNS, can inhibit GARU-GID1A interaction by phosphorylating GARU. So, GNS treatment inhibits TAGK2 kinase activity and releases its inhibition effect on GARU, and then released GARU proteins promote GID1 ubiquitination and degradation. Thus, DELLA proteins are accumulated and GA response is repressed. The manuscript identified two key factors in GID1 stability regulation, which is important and interesting.

Generally, this is a nice story. Here are the comments.

We thank Reviewer #1 for recognizing our proposed model as insightful and interesting. Furthermore, we are grateful for constructive suggestions which have greatly improved our manuscripts.

Comment 1.

1. About how GA regulates GID1 protein levels:

(a) It is interesting that GA treatment inhibits GID1 ubiquitination, and thus increases GID1 protein level (Figure 1d). An in vivo assay in Arabidopsis should be added to confirm this. The authors can also check whether GA induced GID1 accumulation is dependent on the level of E3 GRAU. For example, authors may treat Col and garu mutant with GA and PAC, and then check GID1 protein levels.

Answer 1a.

According to the reviewer's comment, we investigated whether GA-dependent GID1 degradation *in vivo* is suppressed by GA treatment. We analyzed the function of GARU in detail and found that GARU controls the protein level of GID1s in seeds. Although GID1s protein was not observed in dry seeds, it was found after exposing seeds to water (Figure 7a). Furthermore, we show that GID1s protein level decreased under salt stress conditions in a GARU-dependent manner (Figure 7e). We also show that salt-induced GID1 degradation is completely suppressed by exogenous GA₃ (Figure 7f). Through *in vitro* experiments, we show that cell and seeds analysis data support our hypothesis that GA blocks GARU-dependent GID1 degradation. We included these data in Figure 7 and discussed these findings in the revised manuscript (lines 340-365, 386-394 and 453-463)

(b) Figure 5c showed that GA has no effect on GID1-GARU interaction. So, it is possible that GA induced GID1-DELLA interaction block the key lysine residues in the GID1. Can this be tested and discussed?

Answer 1b.

Since 17 lysine residues were found in the amino acid sequence of GID1A, we prepared single, double, or quadruple arginine substitution variants and carried out *in vitro* ubiquitination assays to identify ubiquitination site(s). However, all of these mutants were ubiquitinated by GARU (Supplementary Figure 2), suggesting that ubiquitination occurs on multiple lysine residues. Although GARU could interact with the GA-GID1A-DELLA complex, GID1A ubiquitination was minimally observed. Therefore, ubiquitination site(s) of GID1A is may be the interaction surface with DELLA protein. However, we will have to conduct further studies to reveal the structure of the GA-GID1-DELLA complex.

Comment 2.

2. It will be better to confirm the shifted bands in figure 1g are “Ub-GARU” and “Ub-GID1” using anti-Ub antibody. Does the result in figure1g mean that GARU E3 ligase can be self-ubiquitinated?

Answer 2.

In accordance with the reviewer's comment, we performed the *in vitro* ubiquitination assay again, using a denaturing affinity-precipitation method with HA-tagged ubiquitin and GARU-dependent ubiquitination of GID1A, GID1B, and GID1C proteins are shown by immunoblot with an anti-HA antibody (Figure 1f).

We observed that GARU has self-ubiquitination activity. However, these data are not shown in our manuscript.

Comment 3.

3. How important is GRAU in plant development? In garu mutants growing under mock or normal conditions, GID1 and RGA proteins didn't change and no obvious phenotype was observed (Figure 2e, 6a, 6c-d). GRAU-OX line may be included to check the GID1 protein levels and plant phenotype.

Answer 3.

To answer the reviewer's comment "How important is GRAU in plant development?", we investigated the function of GARU in detail. As described in Answer 1, we found that GARU controls the protein level of GID1s in the seeds. A previous study has indicated that GID1s protein levels were increased by after-ripening or imbibition of the seeds, but not as a result of upregulation of mRNA levels¹, suggesting that the GID1 protein level could be regulated at the translational step or by post-translational modification. The eFP Browser shows that expression of *GARU* gene is detectable in all growth stages (Supplementary Fig. 10). Interestingly, however, it increases at the late dry maturation stages of seeds and decreases after imbibition of water. From these findings, we anticipated that GARU controls the protein level of GID1s in the seeds, and so we investigated this possibility. By phenotypic analysis of Col-0 and a *garu* mutant, we uncovered that GID1s protein is degraded by GARU in the stage of seed maturation and that GARU negatively regulates seed germination (Figure 7a,b). Furthermore, this regulation mechanism is disrupted by imbibition of water. Our studies further revealed that GARU negatively regulates the amount of GID1s protein under salt stress conditions (Figure 7c-h) and decreases sensitivity to GA (Figure 7h). Therefore, GARU acts as a negative regulator to mediate the suppression of seed germination in dry seeds and under salt stress through the degradation

of GID1 protein. We added the above results (lines 340-365: and Figure 7) and discuss these implications (lines 386-394, 453-463) as a new paragraph. We tried to establish a GRAU-OX line, but were unable to do so by the general Agrobacterium-mediated transformation method. However, an orthologue of GARU is apparent in the rice genome and the rice full-length *GARU* cDNA overexpressed in Arabidopsis lines (rice FOX Arabidopsis lines) produced by RIKEN groups had provided OsGARU overexpression lines (<http://ricefox.psc.riken.jp/database/index.php?contents=detail&line=K10643>). However, the RIKEN groups did not provide this strain to us because of the inability to maintain overexpressing lines. This may be because overexpression of GARU gene promotes degradation of GID1 protein, causing germination to be dramatically suppressed.

Comment 4.

4. In Figure 3, ERF13 is not related to the major pathway studied, which only provides clue for identifying TAGK2 as the kinase of GARU. TAGK3 showed phosphorylation activity toward ERF13 (figure 3e), while TAGK2, instead of TAGK3, showed phosphorylation activity toward GARU (figure 4b), which are different. So, I suggest the authors to move figure 3 into supplemental materials. Additionally, the experiments in figure 3g should be done in the same gel to prove that M224A and C227A mutations actually repress TAGK3 kinase activity.

Answer 4.

In accordance with the reviewer's comment, we moved most of the data included in Figure 3 to Supplementary Figure 4. However, we have displayed the following data in Figure 3: (a) Structures of GNS, (b) *in vitro* inhibition assay of TAGKs by GNS, and (c) Tyr phosphorylation analysis of ERF13 in protoplasts. Our manuscript also includes biological and biochemical functional analysis of protein kinases catalyzing Tyr phosphorylation in plants, and our biochemical analyses are the first evidence that plants have protein kinases for Tyr phosphorylation which are target of GNS. In plants, a large number of Tyr-phosphorylated proteins have been identified using proteomics⁴⁹. Unfortunately, although the importance of the role of Tyr phosphorylation signaling has been revealed in

animal studies, biological and physiological roles of Tyr phosphorylation in plants have been unclear. Therefore, we believe that our study will be useful for many researchers.

Comment 5.

5. Figure 4h, why do non-phosphorylation form of Y321F and the two phosphorylation-mimic forms of Y321D and Y321E all partially rescue the GID1A expression? They should show opposite effects, right?

Answer 5.

We apologize for the misunderstanding, which is a result of an accuracy problem in our experiment. As the reviewer points out, in the previous version of the manuscript, it seemed that the amount of GID1A protein increased with co-expression with GARU^{Y321F} mutants compared with wild-type GARU. However, recovery of GID1A protein by GARU^{Y321F} co-expression is possibly a result of the low expression level of GARU^{Y321F} protein compared with wild-type GARU. Thus, we evaluated the amount of GID1A protein in protoplasts by expressing all GARU protein to the same level. As a result, compared with wild-type GARU, the two phosphorylation-mimics GARU^{Y321D} and GARU^{Y321E}, but not GARU^{Y321F}, partially rescued the GID1A expression. We changed this data in Figure 4h.

Comment 6.

6. Figure 4j, in TAGK2WT-OX1 and garu-1 without GNS treatment, the GID1A/C level should increase comparing to Col-0. Why did GID1 protein levels in this plants decrease comparing Col-0? Why is TAGK2KD-OX1 still regulated by GNS?

Answer 6.

To answer the reviewer's comment, we investigated the transcription level of the GID1s gene by RT-qPCR. In TAGK2WT-OX1 and garu-1, the transcription levels of the GID1A gene were slightly decreased compared with wild-type Col-0 (Supplementary Figure 8). Previous study has suggested that to maintain homeostasis of GA signal, excess GA signal causes suppression of GID1s gene expression^{2,3}. Therefore, the transcription levels of three GID1 genes were likely suppressed in TAGK2WT-OX1 and garu-1. However, TAGK2WT-OX1 and garu-1 were not observed GNS-induced GID1 degradation (Figure

4j). Furthermore, these showed high GA sensitivity (Figure 6), suggesting that GID1s protein levels were stabilized in TAGK2WT-OX1 and *garu-1*.

Conversely, TAGK2KD-OX1 showed GNS-induced GID1 degradation (Figure 4j). This transgenic plant expresses both an endogenous TAGK2 and exogenous TAGK2KD genes. However, it cannot suppress the activity of GARU in a Tyr phosphorylation-dependent manner because TAGK2KD protein has no kinase activity. Therefore, TAGK2 is the only endogenous kinase is functioning in TAGK2KD transgenic plants and degradation of GID1 protein occurs after GNS treatment. It is also possible that TAGK2KD protein reduces the amount of functional GNS in cells by capturing GNS. Since GNS sensitivity of TAGK2 is very high (IC₅₀ = 2.56 nM) (Figure 3b), its effect is observed by adding a relatively high concentration (1 μM) of GNS.

Comment 7.

7. Figure 6a, why RGA didn't increase in *tagk2* mutant? Figure 6b show this pattern, but the samples are not in the same gel.

Comment 7a. Figure 6a, why RGA didn't increase in *tagk2* mutant?

Answer 7a.

To answer the reviewer's comment, we analyzed the stability of GID1s protein in each variant. As a result of repeating independent experiments four times, contrary to our expectation, *tagk2* mutant plants showed decreased GID1 protein levels following treatment with GNS (Figure 4j). Furthermore, RGA stability and GA sensitivity were similar to wild-type Col-0 (Figure 6). These results suggest that *TAGK2*-deficiency does not affect stabilization of GID1s or RGA proteins. However, *TAGK2*-overexpression induced increased degradation of DELLA by stabilizing GID1A (Figure 6c). From these results, we propose the following hypothesis: 1) GARU is phosphorylated by TAGK2 as well as other TAGKs and 2) specific, but currently unknown conditions are required for activation of TAGK2. We added the above information to the Discussion section (lines 415-419).

Comment 7b. Figure 6b show this pattern, but the samples are not in the same gel.

Answer 7b.

In accordance with the reviewer's comment, we showed all the samples on the same gel in Figure 6b.

Comment 8.

8. Is it possible to do some genetic relationship analysis of GID1, GARU and TAGK2?

Answer 8.

In order to understand the signal transduction by post-translational modification of proteins, we think that it is important to analyze the biochemical and biological functions of each protein in detail. Therefore, one of the main features of this study is to clarify the regulatory mechanism of GA signaling through GID1 degradation at the molecular level. In this study, our biochemical and biological analysis revealed that GA signaling is both negatively and positively regulated by GARU-dependent GID1A ubiquitination and TAGK2-dependent Tyr phosphorylation of GARU. Since TAGK2 is a negative regulator of GARU, it is expected that the *tagk2/garu* double mutant will show the same phenotype as the *garu* mutant. Conversely, the role of GID1 is dominant in GA signaling, and, in fact, deficiency of the GID1s gene causes many phenotypic abnormalities, such as growth arrest. Therefore, we believe that crossing *gid1* to *garu* or *tagk2* mutants is not an effective approach in gene function analysis because the effect of *garu* or *tagk2* gene deficiency will be hidden by the *gid1* gene deficiency.

Minor :

1. The abbreviation in figures should be well illustrated in figure legends. For example, in Figure 3, "pY" in ERF13 (pY) should be explained.

Answer 1.

In accordance with the reviewer's comment, we have expanded abbreviations in the figure legends.

2. In figure 4j and figure 6b, error bars should be labeled in the quantitative statistics results.

Answer 2.

In accordance with the reviewer's comment, we performed the GID1 stability assay and RGA stability assay, and these assays were repeated independently at least three independent. Signal values of GID1 or RGA proteins and RbcL were measured using ImageJ software. GID1 or RGA protein levels were normalized to RbcL and the value of the control (without GNS-treatment) or starting point was set to 100%. We have indicated that error bars for these data are standard deviations. The two modified graphs are shown in Figure 4j and 6c.

3. There are several spelling errors in the manuscripts, for example, "RAG" in line 515, 516 and 518 should be RGA, "constrictions" in 533 should be constructions.

Answer 3.

In accordance with the reviewer's comment, we fixed the spelling mistakes in our manuscript. "RAG" in lines 515, 516 and 518 have been revised to "RGA" in lines 528, 529 and 532 and "constrictions" in line 533 has been revised to "constructions" in lines 555.

4. The authors should go through the MS to make sure that all the descriptions are clear. For example, in line 315, "To observe the stabilization of GID1A on Tyr321 phosphorylation of GARU", may be changed to "To observe the stabilization of GID1A under the control of Tyr321 phosphorylation of GARU".

Answer 4.

In accordance with the reviewer's comment, we modified "To observe the stabilization of GID1A on Tyr321 phosphorylation of GARU" in line 315 to "To observe the stabilization of GID1A under the control of Tyr321 phosphorylation of GARU" in lines 278-279. Furthermore, we have employed an English editing service to proofread manuscript.

References

1. Hauvermale, A. L., Tuttle, K. M., Takebayashi, Y., Seo, M. & Steber, C. M. Loss of *Arabidopsis thaliana* Seed Dormancy is Associated with Increased Accumulation of

- the GID1 GA Hormone Receptors. *Plant Cell Physiol.* **56**, 1773-1785 (2015).
2. Zentella, R. *et al.* Global analysis of della direct targets in early gibberellin signaling in Arabidopsis. *Plant Cell* **19**, 3037-3057 (2007).
 3. Griffiths, J. *et al.* Genetic characterization and functional analysis of the GID1 gibberellin receptors in Arabidopsis. *Plant Cell* **18**, 3399-3414.

Reviewer #2 (Remarks to the Author):

This manuscript describes the novel finding that the gibberellin (GA) receptor, GID1, is subject to ubiquitination-mediated degradation. The authors used an AlphaScreen-based protein-protein interaction screen to identify the RING E3 ubiquitin ligase (GARU) responsible for GID1 ubiquitination in Arabidopsis and showed that its interaction with GID1 was attenuated by phosphorylation. Furthermore, they identified the tyrosine kinase involved in this phosphorylation, named TAGK2, and demonstrated that its activity is inhibited by genistein, a known kinase inhibitor. Gibberellin promotes plant growth by initiating the degradation of the growth-repressing DELLA proteins in association with GID1. It has previously been shown that genistein stabilised DELLAs, but no evidence could be obtained to suggest that DELLA stability was influenced by its phosphorylation. This manuscript provides an explanation for this observation: by inhibiting the kinase and activating the E3 ligase, genistein destabilises GID1 and thereby stabilises DELLA. This hypothesis is well supported by a series of elegant experiments employing an impressive array of resources.

We thank the reviewer #2 for recognizing the importance of our findings and robustness of our assays. We highly appreciate the reviewer's constructive suggestions to improve our manuscript.

Comment 1.

Indeed the Methods section is extremely long. Could some of this material be provided as on-line only?

Answer 1.

In accordance with the reviewer's comment, we moved the descriptions of methods that "Prediction of AtTAGK3-GNS binding poses", "Constrictions of plasmids for in vitro transcription, transient expression assay and agrobacterium-mediated transformation" and "Multiple sequence alignment" to the Supplementary methods. However, we believe that detailed explanation of our method is important for many researchers to understand and consider. We have made sure that the Methods section is within the word limit for the journal.

Comment 2.

The authors might comment on the physiological significance of this mechanism, which would appear to have a fairly small influence, at least under normal conditions. Despite IC₅₀ values for genistein against TAGK in the low nM range in vitro, 10 μM was used to inhibit Arabidopsis hypocotyl elongation. While 1 μM genistein treatment accumulated DELLA in seedlings, this concentration reduced GID1A/C content only in the presence of cycloheximide. Loss of GARU or overexpression of the TAGK2 kinase had little effect on hypocotyl length, except when seedlings were treated with GA. It would appear that GID1 is protected from ubiquitination by GARU when in association with DELLA, which may explain why GA, which promotes the GID1-DELLA interaction, stabilises GID1 (Fig. 1d). However, GA promotes DELLA degradation and is known to suppress GID1 gene expression through this DELLA destabilising. Thus the mechanism described in the manuscript forms part of the complex homeostasis in GA biosynthesis/signal transduction. These aspects are not clearly explained in the manuscript.

Answer 2.

In accordance with the reviewer's comment, and based on many previous studies and our data of biochemical and physiological analysis, we modified the text in the Discussion (lines 422-441 and 445-468).

Comment 3.

The statistical methodology, which is critical to the work, is not clearly described.

Answer 3.

In accordance with the reviewer's comment, we modified the description of the statistical analysis method in the figure legends of Figure 1, Figure 4, Figure 5, Figure 6, and Figure 7.

Comment 4.

The section on the modelling of the genistein interaction with TAGK2 is something of a diversion and, although of interest, might be omitted if space is limited.

Answer 4.

In accordance with the reviewer's comment, we moved most of the data in Figure 3 to Supplementary Figure 4. However, we have displayed the following data in Figure 3: (a) structures of GNS, (b) *in vitro* inhibition assay of TAGKs by GNS and (c) Tyr phosphorylation analysis of ERF13 in protoplasts. Our manuscript also has the aspect of biological and biochemical functional analysis of protein kinases catalyzing Tyr phosphorylation in plants, and our biochemical analyses are the first evidence that plants have protein kinases for Tyr phosphorylation which are target of GNS. In plants, a large number of Tyr-phosphorylated proteins have been identified using proteomics¹. Unfortunately, although the importance of the role of Tyr phosphorylation signal has been revealed in animal studies, biological and physiological roles of Tyr phosphorylation in plants have been unclear. Therefore, we believe that our study will be useful for many researchers.

Comment 5.

On page 3, line 49: DELLAs do not only act as negative regulators of transcription. In fact, they commonly act as transcriptional co-activators. Sequestration of transcription factors, as described on line 54, is only one mode of action.

Answer 5.

In accordance with the reviewer's comment, we modified the description of the main role of DELLA in the GA signal (lines 47-49 and 53-55).

Reference

1. Zentella, R. *et al.* Global analysis of della direct targets in early gibberellin signaling in *Arabidopsis*. *Plant Cell* **19**, 3037-3057 (2007).

Reviewer #3 (Remarks to the Author):

Sawasaki reports an experimental analysis of the molecular connection between GARU, Genistein and GID1. In my opinion there is not enough novelty in the manuscript and it is not clear the importance of the elucidation of this molecular connection.

Furthermore, the computational section of the paper is completely unclear. It is not clear if they applied docking calculations and which is the pose selection criteria. Finally, a molecular dynamic simulation of the ligand-protein complex should be carried out in order to support the suggested ligand pose.

We are grateful for the reviewer's constructive suggestions, which have allowed us to improve the quality of our manuscript.

Comment 1.

Sawasaki reports an experimental analysis of the molecular connection between GARU, Genistein and GID1. In my opinion there is not enough novelty in the manuscript and it is not clear the importance of the elucidation of this molecular connection.

Answer 1.

The phytohormone group gibberellins (GAs) are diterpene compounds which control a wide range of growth and development phenotypes¹, such as regulating growth and influencing various developmental processes, including stem elongation, germination, dormancy, flowering, sex expression, enzyme induction, and leaf and fruit senescence.

Therefore, to better understand plant physiology, it is extremely important to clarify GA signaling.

There is a direct correlation between GA signaling and endogenous GA levels in plants. Accumulation of GA activates GA signaling by promoting the GID1-dependent degradation of DELLA. However, DELLA induces the expression of GA biosynthesis genes and GID1 genes, and excessive accumulation of GA causes a decrease of endogenous GA and GID1 proteins^{2,3}. These mechanisms are one of the important feedback systems controlling GA signaling. In addition to the mechanism controlling GA synthesis/metabolism and gene expression, previous studies have suggested that plants have multiple mechanisms to promote or suppress GA signaling by regulating the function of DELLA, such as Ser/Thr phosphorylation^{4,5}. However, post-translational modifications of GID1 have not yet been reported. Importantly, a previous study has indicated that GID1s protein levels were increased by after-ripening or imbibition in the seeds, but not at the mRNA level⁶, suggesting that the GID1 protein level could be regulated at the translational step or by post-translational modification.

Because a large amount of GNS is included in soybeans and soy products, studies of GNS function are mainly concerned with the effects on humans and other mammals rather than plants. These studies show that GNS is absorbed in mammals and works as a phytoestrogen and PTK inhibitor^{7,8}. In plants, GNS treatment dramatically inhibits DELLA degradation^{9,10}. Although Hussain et al. (2007) used phenylalanine mutants of Tyr residues in DELLA with Tyr-kinase inhibitor treatments, they found no Tyr phosphorylation of DELLA protein¹¹, suggesting that other proteins in GA signaling are targets of Tyr phosphorylation. However, it is unclear whether Tyr phosphorylation signaling cascades occur in plants because no PTK homolog genes have been found in Arabidopsis and rice genomes^{11,12}.

Based on the above findings of previous studies, we investigated the relationship between GNS and GA signaling and we found that GARU and TAGK are factors linking GNS and GA signaling. Furthermore, we uncovered the GA signaling is both negatively and positively regulated by GARU-dependent GID1A ubiquitination and TAGK2-dependent Tyr phosphorylation of GARU. Our manuscript also contains biological

and biochemical functional analysis of protein kinases catalyzing Tyr phosphorylation in plants and our data are the first evidence to demonstrate the biological significance of trans Tyr-phosphorylation in plants.

Furthermore, our study offers the following new findings:

1. We identified an E3 ubiquitin ligase for the GA-receptor GID1, GARU.
2. In the seedling, GARU plays the role of negative regulator for GA signaling by destabilizing of free GID1 proteins. Furthermore, in the seed, GARU acts as a negative regulator to mediate the suppression of seed germination under salt stress through the degradation of GID1 protein.
3. TAGK2-dependent Tyr phosphorylation acts as an inhibitory switch by preventing GARU-GID1 complex formation, and consequently promotes the stabilization of GID1s protein.
4. TAGK2 is a target of GNS in plant protein kinases.
5. GNS-induced DELLA stabilization is caused by the following three steps: 1) GARU is activated by GNS via inactivation of TAGK2, 2) subsequently, GARU induces ubiquitin-dependent degradation of GID1s, and 3) DELLA is stabilized by reduction of GID1s protein levels in the cell.
6. The GA-induced GID1-DELLA complex formation is more predominant than GARU-dependent GID1 degradation.

Comment 2.

Furthermore, the computational section of the paper is completely unclear. It is not clear if they applied docking calculations and which is the pose selection criteria. Finally, a molecular dynamic simulation of the ligand-protein complex should be carried out in order to support the suggested ligand pose.

Answer 2.

We agree with the reviewer and apologize for the inaccurate description. We modified the description of the method of the computational section and figure (please see Supplementary method and Supplementary Figure 6). Other reviewers recommended

moving to a supplementary figure or removing it because the section on the modelling of GNS-TAGK is of low relevance to the main points in this study. A molecular dynamic analysis is also important for our study, but we agree with the above recommendation and moved the results to Supplementary Figure 6 without doing additional experiments.

References

1. Hedden, P. & Sponsel, V. A Century of Gibberellin Research. *J. Plant Growth Regul.* **34**, 740-760 (2015).
2. Zentella, R. *et al.* Global analysis of della direct targets in early gibberellin signaling in Arabidopsis. *Plant Cell* **19**, 3037-3057 (2007).
3. Middleton, A. M. *et al.* Mathematical modeling elucidates the role of transcriptional feedback in gibberellin signaling. *Proc. Natl Acad. Sci. USA* **109**, 7571-7576 (2012).
4. Dai, C. & Xue, H. W. Rice early flowering1, a CKI, phosphorylates DELLA protein SLR1 to negatively regulate gibberellin signalling. *EMBO J.* **29**, 1916-1927 (2010).
5. Qin, Q. *et al.* Arabidopsis DELLA protein degradation is controlled by a type-one protein phosphatase, TOPP4. *PLoS Genet.* **10**, e1004464 (2014).
6. Hauvermale, A. L., Tuttle, K. M., Takebayashi, Y., Seo, M. & Steber, C. M. Loss of Arabidopsis thaliana Seed Dormancy is Associated with Increased Accumulation of the GID1 GA Hormone Receptors. *Plant Cell Physiol.* **56**, 1773-1785 (2015).
7. Hussain, A., Cao, D. & Peng, J. Identification of conserved tyrosine residues important for gibberellin sensitivity of Arabidopsis RGL2 protein. *Planta* **226**, 475-483 (2007).
8. Dixon, R. A. & Ferreira, D. Genistein. *Phytochemistry* **60**, 205-211 (2002).
9. Qin, Q. *et al.* Arabidopsis DELLA protein degradation is controlled by a type-one protein phosphatase, TOPP4. *PLoS Genet.* **10**, e1004464 (2014).
10. Fu, X. *et al.* Gibberellin-mediated proteasome-dependent degradation of the barley DELLA protein SLN1 repressor. *Plant Cell* **14**, 3191-3200 (2002).
11. Lapcik, O., Honys, D., Koblowska, R., Mackova, Z., Vitkova, M. & Klejdus, B. Isoflavonoids are present in Arabidopsis thaliana despite the absence of any homologue to known isoflavonoid synthases. *Plant Physiol. Biochem.* **44**, 106-114 (2006).

12. Yamada, Y. & Sato, F. Tyrosine phosphorylation and protein degradation control the transcriptional activity of WRKY involved in benzylisoquinoline alkaloid biosynthesis. *Sci. Rep.* **6**:31988 (2016).

Reviewer #4 (Remarks to the Author):

GA plays the important roles in the regulation of plant growth and development throughout the plant life-cycle, including seed germination, leaf expansion, stem elongation and floral development. It is known that GA promotes plant growth via the GA receptor GID1-mediated degradation of DELLA proteins, and previous studies have shown that Ser/Thr phosphorylation/de-phosphorylation of the GA-DELLA regulatory system is involved in GA-induced DELLA degradation. However, the molecular mechanism of this modification still remains unclear. In this manuscript, the authors found that GARU E3 ligase was required for 26S-dependent degradation of GID1. Loss-of-function garu mutant plants displayed higher levels of GID1, and consequently resulting in the degradation of DELLA proteins. The authors also found that Tyr-kinase TAGK2 could inhibit the interaction between GARU and GID1 by phosphorylation of GARU, and the up-regulation of TAGK2 also exhibited the similar GA-sensitive phenotypes as the Arabidopsis garu mutant. The topic of this manuscript is sufficient general interest for researchers follow trends and important development in understanding GID1-mediated GA responses, and I think the authors should give attention to the following points and suggestions.

We thank the reviewer for recognizing our proposed model as an insightful and interesting one. Furthermore, we are grateful for constructive suggestions which have improved our manuscripts.

Comment 1.

1, DELLA proteins function as plant growth repressor in a dosage-dependent manner. The previous studies have shown that GID1-mediated growth responses is not only dependent on the degradation of DELLA proteins, but also dependent upon the activity of DELLA

proteins. For example, the up-regulation of GID1 could promote growth of the *sly1-10* mutant. In Fig.1, the authors found that GNS treatment inhibited plant growth, and concluded that GNS specifically induces the instability of GID1 proteins and consequently provides stabilization of DELLA proteins. Indeed, the authors should provide more evidence to consummate his argument. It is interesting to know more about differences among wild-type plant, *sly1-10* mutant and *della* mutant (lacking five DELLA proteins). Does GNS treatment affect the hypocotyl length of Arabidopsis *gid1 gid1b gid1c* triple mutant?

Answer 1.

We agree with the reviewer that our data is not sufficient to conclude the GNS “specifically” induces the instability of GID1 proteins and consequently provides stabilization of DELLA proteins. Therefore, we deleted “specifically” from the corresponding sentence (line 116).

In accordance with the reviewer’s comment, we analyzed the GNS-induced growth inhibition in the DELLA mutant (lacking five DELLA proteins) and the results are shown in Supplementary Figure 1.

As the reviewer suggests, accumulation of DELLA may be caused by decrease or inactivation of GA signaling factors, such as GID1 or F-box protein SLY1. Indeed, previous studies have shown that the deficiency of *GID1* and *SLY1* genes dramatically promote DELLA accumulation and suppresses GA signal. Thus, the functions of GID1 and SLY1 are dominant in GA signaling. However, in this study, the GNS-induced DELLA accumulation and growth inhibition were not dramatic. Therefore, it is difficult to analyze the influence of GNS in *sly1-10* and *gid1a/gid1b/gid1c* mutants. In consideration of the possibility that GNS may act on other proteins, such as SLY1, we added “one of the mechanisms of GNS-induced DELLA accumulation” to the Discussion section (lines 428-429).

Comment 2.

2, the authors identified two RING-type E3 ubiquitin ligases, *at2g40830* and *at1g4750*, that interacted strongly with Arabidopsis GID1A. The authors performed in vitro ubiquitination assays and found that all Arabidopsis GID1 proteins (*AtGID1A*, *AtGID1B* and *AtGID1C*)

could be ubiquitinated by the gene product of at2g40830, but ubiquitinated AtGID1A proteins were not detected when AtGID1 was incubated with the gene product of at1g4750. The results of this analysis did not conclude that both AtGID1B and AtGID1C could not be ubiquitinated by the gene product of at1g4750. Thus, the authors should performed *in vitro* ubiquitination assays using the gene product of at1g4750.

Answer 2.

In accordance with the reviewer's comment, we performed the *in vitro* ubiquitination assay again using a denaturing affinity-precipitation method with HA-tagged ubiquitin. GARU-dependent ubiquitination of GID1A, GID1B, and GID1C proteins were detected by immunoblot with an anti-HA antibody. As a result, GARU could ubiquitinate GID1A, GID1B, and GID1C. In contrast, at1g47579 protein could not ubiquitinate all GID1s protein. We show this result in Figure 1f.

Comment 3.

3, to confirm GARU-mediated degradation of GID1, the authors should analyzed the accumulation of Arabidopsis GID1 proteins and their ubiquitination using single and double mutants of at2g40830 and at1g4750.

Answer 3.

The *in vitro* ubiquitination assay showed that at1g47579 protein could not ubiquitinate all GID1s proteins. Therefore, we believe that analysis of at1g47579 mutant was not important in this study.

Comment 4.

4, in Fig. 2d, the image of YFP-GARU is not clear. It is interested to know the localization of interaction between GID1A and GARU proteins using BiFC assays. Does GA treatment influence their interaction and/or localization of GARU?

Answer 4.

In accordance with the reviewer's comment, we performed the GARU-GID1A interaction analysis in the cells using the BiFC assay. GARU-GID1A interaction was observed in the nucleus in the presence and absence of GA₃, indicating that GARU could interact with

GID1 with or without GA₃ in the nucleus. The result of the BiFC assay is shown in Figure 2e. Furthermore, we changed the image of YFP-GARU in Figure 2d.

Comment 5.

5, as described above, the GA-GID1-DELLA regulatory system plays the important roles in the control of plant growth and development, including seed germination, root extension, hypocotyl elongation and flowering time etc. In this manuscript, the authors only mentioned hypocotyl length. How about other growth phenotypes?

Answer 5.

To answer the reviewer's comment, we investigated the function of GARU in detail. We found that GARU controls the protein level of GID1s in the seeds. A previous study has indicated that GID1s protein levels were increased by after-ripening or imbibition in the seeds, but not at the mRNA level¹, suggesting that the GID1 protein level could be regulated at the translational step or by post-translational modification. The eFP Browser shows that expression level of *GARU* gene is observed in all growth stages (Supplementary Figure 10). Interestingly, however, it increases at the late dry maturation stages of seeds, and decreases after imbibition of water. From these findings, we anticipated that GARU controls the protein level of GID1s in the seeds, and so we investigated this possibility. By phenotypic analysis of Col-0 and *garu* mutants, we uncovered that GID1s protein is degraded by GARU in the stage of seed maturation, and GARU negatively regulates seed germination (Figure 7a,b). Furthermore, this regulation mechanism is disrupted by imbibition of water. Our studies further revealed that GARU negatively regulates the amount of GID1s protein under the salt stress (Fig. 7c-h) and decreases sensitivity to GA (Fig. 7h). Taken together, our data suggest that GARU acts as a negative regulator to mediate the suppression of seed germination in dry seeds and under the salt stress through the degradation of GID1 protein.

We added the above results (line2 340-365 and Figure 7) and discussion (lines 386-394 and 456-468) as a new paragraph.

Reference

1. Hauvermale, A. L., Tuttle, K. M., Takebayashi, Y., Seo, M. & Steber, C. M. Loss of *Arabidopsis thaliana* Seed Dormancy is Associated with Increased Accumulation of the GID1 GA Hormone Receptors. *Plant Cell Physiol.* **56**, 1773-1785 (2015).

Reviewers' comments:

Reviewer #1 (Remarks to the Author):

The MS of this version has been improved, on both biochemical and biological roles of the components in TAGK2-GARU-GID1 module. I still have the following concerns.

(1) The bands of the western blot in Fig. 7, including both GID1A/C and CBB, are faint. The authors need to improve the quality by changing contrast ratio and brightness, or redoing some of them. This figure shows the regulations in plant, which is important for clarifying the biological significance of the regulation module identified in this MS. In Fig. 7f, "α-GID1A" should be revised to "α-GID1A/C".

(2) In Fig. 7f&h, the authors should explain why germination rate of garu-1 increased with NaCl and GA treatment. GID1A/C protein level in garu-1 didn't increase in this condition compared to Col-0 (Fig. 7f). For Col-0, germination rate under NaCl+GA3 is even lower compared to under NaCl at 2 days. This is strange because GA3 usually promote germination.

In Fig7h, in order to evaluate the biological role of GARU in salt and GA mediated GID1 degradation, the authors should compare the germination rates between Col-0 and garu-1 mutant under mock, NaCl and NaCl+GA conditions in parallel, and summarize the results in one line chart, instead of in two separated ones.

(3) In the fig 2e, for the BiFC assay, negative controls are necessary.

Reviewer #2 (Remarks to the Author):

In their revision the authors have responded thoroughly to the reviewers' concerns and have introduced new experiments, in particular to address the physiological role of GID1 post-translational regulation via ubiquitination by the GARU E3 ligase. The work described in the manuscript introduces a novel mechanism for regulating gibberellin signalling with some indication of the scenarios where it has physiological relevance. Experiments with the garu-1 knock-out line suggest involvement in seed dormancy/germination and in response to salt stress, at least in the seed. The findings are in line with other mechanisms for regulation of gibberellin signaling that override homeostasis in relation to developmental change and response to stress. The finding that gibberellin stabilises GID1 protein as opposed to its known suppression of GID1 mRNA accumulation is of particular interest. Generally the manuscript is quite well written, but the grammar could be improved in places, particularly where the meaning is unclear.

Examples:

The sentence on lines 98-100 is ambiguous and can be clarified simply by adding respectively at the end.

On lines 106-107, the meaning of the sentence is vague and requires clarification.

Reviewer #3 (Remarks to the Author):

The paper can be accepted as is.

Reviewer #4 (Remarks to the Author):

In this revised manuscript, the authors have added additional experiments and answered the questions which I addressed, now it is acceptable for the publication in this journal.

Response to reviewers

We have enclosed our revised manuscript entitled “**Tyrosine Phosphorylation of GARU RING E3 Ligase for Gibberellin Receptor Regulates Gibberellin Response**” by Nemoto et al. Thank you for your positive comments on our manuscript and your thoughtful critiques. Below, we have addressed point-by-point your constructive critiques, and modified the text accordingly.

Reviewer #1 (Remarks to the Author):

The MS of this version has been improved, on both biochemical and biological roles of the components in TAGK2-GARU-GID1 module. I still have the following concerns.

We thank the reviewer #1 for recognizing the importance of our findings and robustness of our assays. We highly appreciate the reviewer’s constructive suggestions to improve our manuscript.

Comment 1.

The bands of the western blot in Fig. 7, including both GID1A/C and CBB, are faint. The authors need to improve the quality by changing contrast ratio and brightness, or redoing some of them. This figure shows the regulations in plant, which is important for clarifying the biological significance of the regulation module identified in this MS. In Fig. 7f, “ α -GID1A” should be revised to “ α -GID1A/C”.

Answer 1.

In accordance with the reviewer’s comment, we changed the contrast ratio and brightness of the images of immunoblotting, CBB staining and Ponceau S staining in Fig.7 a, c, d-f. In addition, we revised “ α -GID1A” to “ α -GID1A/C” in Fig. 7f. We described in the method section about changing the contrast ratio and brightness of the images by ImageJ software

(lines 509-511 were indicated with yellow highlighted text).

Comment 2.

In Fig. 7f&h, the authors should explain why germination rate of *garu-1* increased with NaCl and GA treatment. GID1A/C protein level in *garu-1* didn't increase in this condition compared to Col-0 (Fig. 7f). For Col-0, germination rate under NaCl+GA3 is even lower compared to under NaCl at 2 days. This is strange because GA3 usually promote germination.

In Fig 7h, in order to evaluate the biological role of GARU in salt and GA mediated GID1 degradation, the authors should compare the germination rates between Col-0 and *garu-1* mutant under mock, NaCl and NaCl+GA conditions in parallel, and summarize the results in one line chart, instead of in two separated ones.

Comment 2a.

In Fig. 7f&h, the authors should explain why germination rate of *garu-1* increased with NaCl and GA treatment. GID1A/C protein level in *garu-1* didn't increase in this condition compared to Col-0 (Fig. 7f).

Answer 2a.

Our studies further revealed that GARU negatively regulates the amount of GID1s protein in seeds under salt stress (Fig. 7c-e,g), and GA₃ dramatically suppresses it (Fig. 7f). However, under salt stress, germination of Col-0 was not rescued to the same level as *garu-1* by GA₃ treatment (Fig. 7h). These results suggested that GID1 protein in *garu-1* is more stable than Col-0 under salt stress. It may be not possible to stabilize GID1s protein over a long period by the GID1-GA-DELLA complex formation under salt stress because DELLA protein is rapidly degraded. We added the above information to the Discussion section (lines 386-394 were indicated with yellow highlighted text).

Comment 2b.

For Col-0, germination rate under NaCl+GA3 is even lower compared to under NaCl at 2 days. This is strange because GA3 usually promote germination.

In Fig7h, in order to evaluate the biological role of GARU in salt and GA mediated GID1 degradation, the authors should compare the germination rates between Col-0 and *garu-1* mutant under mock, NaCl and NaCl+GA conditions in parallel, and summarize the results in one line chart, instead of in two separated ones.

Answer 2b.

In results before the revision, at first glance, germination rate of Col-0 under NaCl+GA₃ is even lower compared to under NaCl at 2 days, but there was no significant difference between them. However, we investigated the cause why GA₃ did not promote germination of Col-0 seeds under the conditions before the revision.

In results before the revision, germination assay was performed with the seeds submerged in NaCl solution. However, low-oxygen stress imposed by waterlogging is probably a serious impediment to plant germination and growth¹. Furthermore, long-term excessive salt stress causes severe damage to seeds, such as disruption to the structure of enzymes and other macromolecules, damage to cell organelles and the plasma membrane, the disruption of respiration and protein synthesis². From these findings, we thought that two excessive stresses, low-oxygen stress and salt stress, suppressed the GA response by activating the survival mechanism of the seeds. Therefore, we examined the experimental conditions of germination assay again. In detail, the stratification treatment at 4 °C was shortened from 4 days to 2 days in salt solution. After stratification, seeds were germinated on solid medium containing with NaCl or NaCl+GA₃. As a result, germination rate of Col-0 under NaCl+GA₃ was higher compared to under NaCl. Furthermore, germination rate of *garu* mutant was higher than that of Col-0 under both NaCl and NaCl+GA₃ conditions. This result supports our hypothesis that GARU acts as a negative regulator and suppresses seed germination under salt stress through the degradation of GID1 protein. Based on the above results, we performed the germination assay under mock, NaCl and NaCl+GA conditions in parallel, and summarized the results in one line chart in Figure 7h. Furthermore, we modified the description of the result (lines 361-364 were indicated with yellow highlighted text).

Comment 3.

In the fig 2e, for the BiFC assay, negative controls are necessary.

Answer 3.

In accordance with the reviewer's comment, we performed the BiFC assay in which nYFP and cYFP (non fusion forms), as negative controls, were co-expressed with GID1A-cYFP or nYFP-GARU. As a result, YFP fluorescence was observed in co-expressed of GID1A-cYFP/nYFP-GARU in the nucleus. In contrast, YFP fluorescence was not observed in co-expressed of nYFP/GID1A-cYFP, nYFP-GARU/cYFP and nYFP/cYFP. These results are shown in Figure 2e.

References

1. Gibbs, D. J. *et al.* Homeostatic response to hypoxia is regulated by the N-end rule pathway in plants. *Nature* 479, 415-418 (2011).
2. Ibrahim, E. A. Seed priming to alleviate salinity stress in germinating seeds. *J. Plant Physiol.* 192, 38-46 (2016).

Reviewer #2 (Remarks to the Author):

In their revision the authors have responded thoroughly to the reviewers' concerns and have introduced new experiments, in particular to address the physiological role of GID1 post-translational regulation via ubiquitination by the GARU E3 ligase. The work described in the manuscript introduces a novel mechanism for regulating gibberellin signalling with some indication of the scenarios where it has physiological relevance. Experiments with the *garu-1* knock-out line suggest involvement in seed dormancy/germination and in response to salt stress, at least in the seed. The findings are in line with other mechanisms for regulation of gibberellin signaling that override homeostasis in relation to developmental

change and response to stress. The finding that gibberellin stabilises GID1 protein as opposed to its known suppression of GID1 mRNA accumulation is of particular interest.

We are grateful for the reviewer's constructive suggestions, which have allowed us to improve the quality of our manuscript.

Comment 1.

Generally the manuscript is quite well written, but the grammar could be improved in places, particularly where the meaning is unclear.

Examples:

The sentence on lines 98-100 is ambiguous and can be clarified simply by adding respectively at the end.

On lines 106-107, the meaning of the sentence is vague and requires clarification.

Answer 1.

In accordance with the reviewer's comment, we modified the description about the functions of GARU and TAGK2 in GA signaling to a simple and clear sentence (lines 92-100 were indicated with yellow highlighted text). Furthermore, we modified the description that inferring the function of PTK in the GA signaling to be clear (lines 106-107 were indicated with yellow highlighted text).

Reviewer #3 (Remarks to the Author):

The paper can be accepted as is.

We thank the reviewer for the positive evaluation and kind help in improving our manuscript.

Reviewer #4 (Remarks to the Author):

In this revised manuscript, the authors have added additional experiments and answered the questions which I addressed, now it is acceptable for the publication in this journal.

We thank the reviewer for the positive evaluation and kind help in improving our manuscript.

REVIEWERS' COMMENTS:

Reviewer #1 (Remarks to the Author):

With the additional experiments and explanations, the manuscript has been further improved and can be accepted now.

Response to reviewer(s)

Reviewer #1 (Remarks to the Author):

With the additional experiments and explanations, the manuscript has been further improved and can be accepted now.

We thank the reviewer#1 for the positive evaluation and kind help in improving our manuscript.